# Toward incompatible quantum limits on multiparameter estimation

Binke Xia [1], Jingzheng Huang [1,2,3] ✉, Hongjing Li[1,2,3], Han Wang[1] & Guihua Zeng [1,2,3] ✉

Achieving the ultimate precisions for multiple parameters simultaneously is an outstanding challenge in quantum physics, because the optimal measurements for incompatible parameters cannot be performed jointly due to the Heisenberg uncertainty principle. In this work, a criterion proposed for multiparameter estimation provides a possible way to beat this curse. According to this criterion, it is possible to mitigate the influence of incompatibility meanwhile improve the ultimate precisions by increasing the variances of the parameter generators simultaneously. For demonstration, a scheme involving high-order Hermite-Gaussian states as probes is proposed for estimating the spatial displacement and angular tilt of light at the same time, and precisions up to 1.45 nm and 4.08 nrad are achieved in experiment simultaneously. Consequently, our findings provide a deeper insight into the role of Heisenberg uncertainty principle in multiparameter estimation, and contribute in several ways to the applications of quantum metrology.

Quantum parameter estimation plays a vital role in a wide range of physics and engineering, including interferometry[1–4], superresolution[5–8], optical sensing[9–11] and so on. Generally, the ultimate precision of estimating unknown parameter is characterized by the quantum Cramér-Rao (QCR) bound[12–14], which can be explicitly calculated by the quantum Fisher information matrix (QFIM)[15,16]. For single-parameter estimation, the QCR bound is always attainable by assigning an optimal measurement[17–19]. In multiparameter estimation, there is a quantum limit (QL) point that the estimating precisions for all parameters achieve their QCR bounds simultaneously. This QL point can be saturated only if the optimal measurements for all parameters are compatible, such as the estimation of multiple phases[20,21] and parameters in SU(2) operators with ancillary qubits[22,23]. Unfortunately, the optimal measurements for incompatible parameters can not be jointly performed due to the Heisenberg uncertainty principle (HUP), thus the QL point is unachievable[24–26]. To determine the attainable precisions of incompatible parameters, a trade-off relation between the estimation inaccuracies for different parameters[27] was recently revealed by incorporating the HUP and Ozawa's uncertainty relation[28,29].

Although the QL point of incompatible parameters is unachievable, we find that their trade-off precision bound can be dramatically improved by devising an appropriate measurement probe. By revealing the connection between variances of estimation errors and generators[30] corresponding to different parameters, a criterion is proposed to guide the design of probe. It significantly indicates that, increasing the variances of generators simultaneously may not only improves the quantum limits, but also mitigates the incompatibility of corresponding parameters, and then the estimating precisions can approach to the QL point asymptotically.

In this work, we study a scheme that measuring the parameters of momentum and position simultaneously in a quantum system. By employing the Hermite-Gaussian (HG) state as the measurement probe, of which the momentum and position variances increase simultaneously along with the mode number, the QL point of this pair of incompatible parameters can be approached asymptotically according to our criterion. To suppress the technical noise[31–33], we also introduce the post-selected weak measurement technique in our system. For demonstration, our scheme is performed in an optical experiment by employing HG beams, where the incompatible

[1]State Key Laboratory of Advanced Optical Communication Systems and Networks, Institute for Quantum Sensing and Information Processing, School of Sensing Science and Engineering, Shanghai Jiao Tong University, Shanghai 200240, China. [2]Hefei National Laboratory, Hefei 230088, China. [3]Shanghai Research Center for Quantum Sciences, Shanghai 201315, China. ✉e-mail: jzhuang1983@sjtu.edu.cn; ghzeng@sjtu.edu.cn

parameters are produced by the weak transverse displacement and angular tilt of light. When the number of HG mode increases, the estimation errors of different parameters decrease and approach to their incompatible quantum limits simultaneously, which agrees well with our theoretical prediction. Especially, precisions up to 1.45 nm and 4.08 nrad for estimating spatial displacement and angular tilt of light are achieved at the same time in our experiment. As the theoretical and experimental results all indicate that our method not only mitigates the incompatibility to approach the simultaneous quantum limits, but also improves the quantum limits to achieve the smaller estimation errors, our findings will be of interest to many application scenarios in quantum metrology, e.g., polarization microscopy[34], qubit tomography[35], estimation of a multidimensional field[36], metrology of correlated phase and loss[37], and joint measurement of phase and phase diffusion[38].

## Results and discussion

### Quantum criterion of multiparameter estimation

Let us start with the quantum multiparameter estimation process illustrated in Fig. 1. An evolution operator $\hat{U}(\boldsymbol{g})$ with $n$ unknown parameters $\boldsymbol{g} = (g_1, g_2, \ldots, g_n) \in \mathbb{R}^n$ transfers a probe state $|\psi\rangle$ to a parameterized state of $|\psi_{\boldsymbol{g}}\rangle$. Afterwards, the values of $\boldsymbol{g}$ can be estimated via performing measurements on $|\psi_{\boldsymbol{g}}\rangle$. By devising the optimal measurements for all parameters, the QCR bound gives the quantum limit on the precision of every parameter via the QFIM.

Generally, the quantum Fisher information (QFI) of parameter $g_i$ can be derived as $\mathcal{Q}_{ii} = 4\langle\Delta\hat{H}_i^2\rangle$ ($\mathcal{Q}_{ii}$ is the $i$-th diagonal element of the corresponding QFIM $\mathcal{Q}$), where $\langle\Delta\hat{H}_i^2\rangle \equiv \langle\psi|\hat{H}_i^2|\psi\rangle - \langle\psi|\hat{H}_i|\psi\rangle^2$ is the variance of $\hat{H}_i$, with $\hat{H}_i = \mathrm{i}[\frac{\partial}{\partial g_i}\hat{U}^\dagger(\boldsymbol{g})]\hat{U}(\boldsymbol{g})$ being the generator of $g_i$, which leads to the quantum limit of parameter $g_i$ given by[30]:

$$(\delta g_i)^2\langle\Delta\hat{H}_i^2\rangle \geq \frac{1}{4\nu} \tag{1}$$

where $\nu$ is the measured samples number. This limit can be saturated via an optimal measurement connected to generator $\hat{H}_i$[17,39] in the single-parameter estimation. Moreover, this limit also implies that the precision of a single parameter can be improved by maximizing the variance of $\hat{H}_i$, which requires to optimize the probe state.

However, achieving the quantum limits for all parameters simultaneously requires their corresponding optimal measurements being compatible, which is granted by checking the weak commutative condition[17]:

$$\mathrm{Im}\left(\frac{\partial\langle\psi_{\boldsymbol{g}}|}{\partial g_i}\frac{\partial|\psi_{\boldsymbol{g}}\rangle}{\partial g_j}\right) = \langle[\hat{H}_i, \hat{H}_j]\rangle = 0 \tag{2}$$

where $[\hat{H}_i, \hat{H}_j] \equiv \hat{H}_i\hat{H}_j - \hat{H}_j\hat{H}_i$, $\langle\cdot\rangle$ denotes for $\langle\psi|\cdot|\psi\rangle$. For two incompatible parameters $g_i$ and $g_j$ who do not satisfy the weak commutative condition, their optimal measurements can not be performed simultaneously, so their quantum limits are incompatible to be achieved. To deeply investigate the incompatibility from the physical insight, we relate the generators $\hat{H}_i$ and $\hat{H}_j$ of the incompatible parameters by

Heisenberg's uncertainty relation:

$$\langle\Delta\hat{H}_i^2\rangle\langle\Delta\hat{H}_j^2\rangle \geq \frac{1}{4}|\langle[\hat{H}_i, \hat{H}_j]\rangle|^2 \tag{3}$$

We will show that in follows, the incompatibility between measurements can be dramatically mitigated by using appropriate probe state, so that the QL point can be approached asymptotically. To achieve this goal, the probe state should be chosen maximizing the variances of $\hat{H}_i$ and $\hat{H}_j$. According to Eq. (1), this method can also improve the quantum limits to achieve smaller estimation errors regarding to parameters $g_i$ and $g_j$.

To illustrate our result, we first define the quantum multiparameter estimation criterion (QMEC) as follow:

$$\mathcal{S}_{ij} = \frac{4\langle\Delta\hat{H}_i^2\rangle\langle\Delta\hat{H}_j^2\rangle}{|\langle[\hat{H}_i, \hat{H}_j]\rangle|^2} \tag{4}$$

According to Eq. (3), we have $\mathcal{S}_{ij} \geq 1$. The value of $\mathcal{S}_{ij}$ corresponds to the uncertainties of $\hat{H}_i$ and $\hat{H}_j$. They are minimized simultaneously when $\mathcal{S}_{ij} = 1$ and become larger when $\mathcal{S}_{ij}$ increases.

With the QMEC in hand, the achievable precision limit in multiparameter estimating is given by a trade-off relation[27] (see the Supplementary Note 1 for derivation):

$$\begin{aligned}2 \quad & -\frac{1}{\nu(\delta\tilde{g}_i)^2} - \frac{1}{\nu(\delta\tilde{g}_j)^2} + 2\sqrt{1 - \frac{1}{\mathcal{S}_{ij}}} \\ & \times\sqrt{\left[1 - \frac{1}{\nu(\delta\tilde{g}_i)^2}\right]\left[1 - \frac{1}{\nu(\delta\tilde{g}_j)^2}\right]} \geq \frac{1}{\mathcal{S}_{ij}}\end{aligned} \tag{5}$$

where $\delta\tilde{g}_i = \sqrt{\mathcal{Q}_{ii}}\delta g_i = 2\sqrt{\langle\Delta\hat{H}_i^2\rangle}\delta g_i$ and $\delta\tilde{g}_j = \sqrt{\mathcal{Q}_{jj}}\delta g_j = 2\sqrt{\langle\Delta\hat{H}_j^2\rangle}\delta g_j$ are the normalized estimation errors for the normalized parameters $\tilde{g}_i = \sqrt{\mathcal{Q}_{ii}}g_i$ and $\tilde{g}_j = \sqrt{\mathcal{Q}_{jj}}g_j$, which are dimensionless and reflect the discrepancies between the practical attainable precisions and the corresponding quantum limits. This trade-off relation not only reveals the achievable precision limit of multiparameter estimation, but also indicates a possible way to achieve simultaneous ultimate precisions via the criterion $\mathcal{S}_{ij}$.

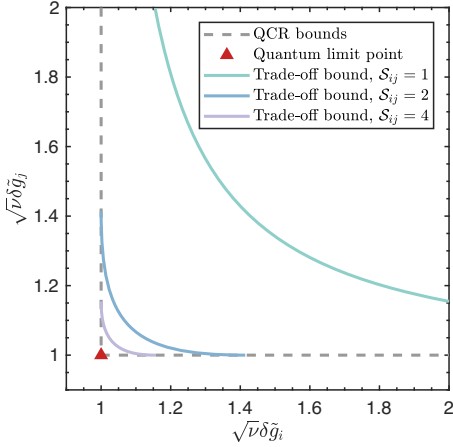

Fig. 2 | **Precision limits of estimating parameters $g_i$ and $g_j$ simultaneously.** The $x$ axis and $y$ axis are separately the normalized estimation errors of parameters $g_i$ and $g_j$. The gray dashed lines are separately the QCR bounds for parameters $g_i$ and $g_j$, which are directly obtained from their corresponding QFI. The cross point (red triangle in figure) of gray dashed lines is the quantum limit point where both parameters achieve the theoretical ultimate precision. The green solid curve stands for the trade-off bound of parameters $g_i$ and $g_j$ with $\mathcal{S}_{ij} = 1$, which corresponds to the minimum-uncertainty probe state. The blue and purple solid curves are separately the trade-off bounds with $\mathcal{S}_{ij} = 2$ and $\mathcal{S}_{ij} = 4$.

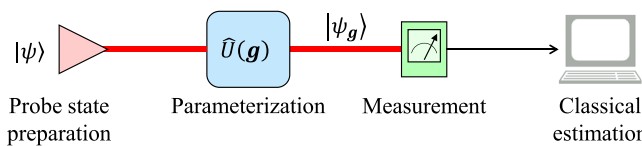

Fig. 1 | **Schematic of the quantum parameter estimation.** A generalized quantum parameter estimation process consists of four steps: probe state preparation, parameterization, measurement and classical estimation.

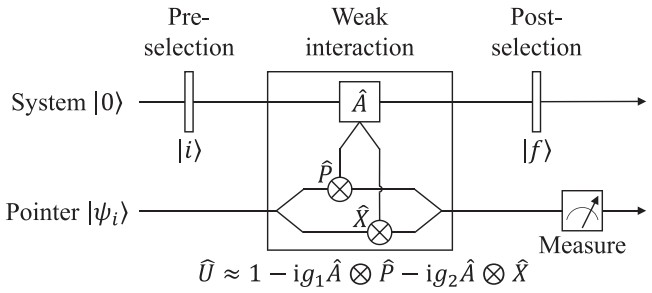

**Fig. 3 | Post-selected weak measurement scheme for the incompatible parameters.** Here we investigate the parameters generated by momentum operator $\hat{P}$ and position operator $\hat{X}$ during the weak interaction procedure.

Based on Eq. (4) and Eq. (5), the trade-off precision bounds are compared with the single parameter limits in Fig. 2. Here, the quantum limit (QL) point represents the simultaneous ultimate precisions for the parameters $g_i$ and $g_j$. It clearly shows that the trade-off precision bound approaches the QL point when $S_{ij}$ increases, and the worst trade-off bound appears when $S_{ij} = 1$. In other words, the larger the uncertainties of $\hat{H}_i$ and $\hat{H}_j$ are, the easier to achieve the ultimate precisions simultaneously.

The QMEC together with Eq. (5) provide us a guidance for optimizing the probe state for multiparameter estimation. As an example, we consider the simultaneous estimations on parameters with generators of momentum operator $\hat{P}$ and position operator $\hat{X}$. In this case, the unitary parameterization is $\hat{U}(\boldsymbol{g}) = \exp(-ig_1\hat{P} - ig_2\hat{X})$, where $g_1$ and $g_2$ representing position displacement and momentum kick are the parameters of interest, and the corresponding QMEC can be easily calculated as $S_{12} = S_{21} = 4\langle\Delta\hat{P}^2\rangle\langle\Delta\hat{X}^2\rangle$. (Without loss of generality, we adopt units making $\hbar = 1$ in this article.) Let us take Gaussian state, which is usually used as probe [42,43] for example. Gaussian state satisfying $\langle\Delta\hat{P}^2\rangle\langle\Delta\hat{X}^2\rangle = 1/4$ yields $S_{12} = 1$, which means it gets the worst trade-off precision bound. In this sense, the high-order Hermite-Gaussian (HG) state[40] with larger uncertainties on $\hat{P}$ and $\hat{X}$ could be a better choice for probe.

## Simultaneous measurement of incompatible parameters

To practical verify our theory, we investigate a post-selected weak measurement scheme where a pair of incompatible parameters is loaded simultaneously during the weak interaction procedure. As is illustrated in Fig. 3, a two-level system and a pointer with continuous degree of freedom are prepared in states of $|i\rangle$ and $|\psi_i\rangle$ respectively. During the weak interaction procedure, the position displacement parameter $g_1$ and momentum kick parameter $g_2$ are coupled simultaneously to pointer with a impulse Hamiltonian $\hat{H}_I = \left(g_1\hat{P}\otimes\hat{A} + g_2\hat{X}\otimes\hat{A}\right)\delta(t - t_0)$, where $\hat{A}$ is an Hermitian operator on the two-level system. Thus, the theoretical unitary parameterization in weak measurement scheme is $\hat{U}(\boldsymbol{g}) = \hat{U}_w = \exp(-i\int\hat{H}_I dt)$. By adopting the weak interaction assumption $g_1\sigma_p \ll 1$ and $g_2\sigma_x \ll 1$, the unitary parameterization can be approximately calculated as

$$\hat{U}(\boldsymbol{g}) = \hat{U}_w \approx 1 - ig_1\hat{A}\otimes\hat{P} - ig_2\hat{A}\otimes\hat{X} \qquad (6)$$

To individually read out the measurement parameters from the pointer, we post-select the system by state $|f\rangle$, and turn the final state in whole to be a product state $|\psi_f\rangle|f\rangle$, where

$$|\psi_f\rangle \approx \left(1 - ig_1 A_w\hat{P} - ig_2 A_w\hat{X}\right)|\psi_i\rangle \qquad (7)$$

is the final state of pointer, and $A_w = \langle f|\hat{A}|i\rangle/\langle f|i\rangle$ is weak value. Then the corresponding QFIM can be obtained as:

$$\mathcal{Q} \approx 4|A_w|^2 \begin{pmatrix} \langle\Delta\hat{P}^2\rangle_i & \frac{1}{2}\langle\{\hat{P},\hat{X}\}\rangle_i \\ \frac{1}{2}\langle\{\hat{X},\hat{P}\}\rangle_i & \langle\Delta\hat{X}^2\rangle_i \end{pmatrix} \qquad (8)$$

where $\langle\cdot\rangle_i$ denotes for $\langle\psi_i|\cdot|\psi_i\rangle$. Thus, the QMEC in our measurement scheme for incompatible parameters $g_1$ and $g_2$ can be calculated as

$$S_{12} = S_{21} = 4\langle\Delta\hat{P}^2\rangle_i\langle\Delta\hat{X}^2\rangle_i \qquad (9)$$

which has the same expression of the post-selection-free scheme, and is only dependent on the momentum and position uncertainty of initial pointer state. (See Supplementary Note 1 for calculation details.)

In the conventional weak measurement scheme, the pointer is usually chosen to have Gaussian distribution, which leads to the worst QMEC $S_{12} = 1$. According to the trade-off relation of parameters' precisions in Eq. (5), the corresponding trade-off bound of Gaussian pointer is given by:

$$\frac{1}{\nu(\delta\tilde{g}_1)^2} + \frac{1}{\nu(\delta\tilde{g}_2)^2} \leq 1 \qquad (10)$$

where

$$\begin{aligned} \delta\tilde{g}_1 &= \sqrt{\mathcal{Q}_{11}}\delta g_1 = 2|A_w|\sqrt{\langle\Delta\hat{P}^2\rangle_i}\delta g_1 \\ \delta\tilde{g}_2 &= \sqrt{\mathcal{Q}_{22}}\delta g_2 = 2|A_w|\sqrt{\langle\Delta\hat{X}^2\rangle_i}\delta g_1 \end{aligned} \qquad (11)$$

are respectively the normalized estimation errors of parameters $g_1$ and $g_2$. For simplicity, we can also define the normalized parameter vector

$$\tilde{\boldsymbol{g}} = (\tilde{g}_1, \tilde{g}_2) = 2A_w\left(\sqrt{\langle\Delta\hat{P}^2\rangle_i}g_1, \sqrt{\langle\Delta\hat{X}^2\rangle_i}g_2\right) \qquad (12)$$

Here, we illustrated the the normalized estimation errors $\delta\tilde{g}_1$ and $\delta\tilde{g}_2$ with Gaussian pointer in Fig. 4a, where the cross point (red triangle in figure) of the QCR bounds (gray dashed lines in figure) is the QL point $\mathcal{P}_Q = \sqrt{\nu}(\delta\tilde{g}_{1Q}^{min}, \delta\tilde{g}_{2Q}^{min}) = (1,1)$. However, according to the trade-off relation in Eq. (5), the achievable boundary of simultaneous estimation errors is dependent on the probe's uncertainty. For Gaussian pointer, the trade-off bound given by Eq. (10) is illustrated by the blue solid curve in Fig. 4a, where improving the estimating precision of one parameter toward the quantum limit will make the estimation error of the other parameter turn to infinite. It means that the QL point can not be achieved for incompatible parameters via Gaussian pointer.

To improve the trade-off bound of the incompatible parameters, we employ the $n$-order Hermite-Gaussian (HG) state as our initial pointer, whose spatial wave function is:

$$\psi_n(x) = \frac{1}{\sqrt{2^n n!}\sqrt{2\pi\sigma_0^2}} H_n\left(\frac{x}{\sqrt{2}\sigma_0}\right)\exp\left(-\frac{x^2}{4\sigma_0^2}\right) \qquad (13)$$

where $\sigma_0^2$ is the variance of spatial distribution of fundamental Gaussian state. From the view of quantum mechanics, $n$-order HG state can be denoted by the $n$-order eigenket of harmonic oscillators $|n\rangle$. Then the momentum and position uncertainty can be calculated as $\langle\Delta\hat{P}^2\rangle_i = (2n+1)/4\sigma_0^2$, $\langle\Delta\hat{X}^2\rangle_i = (2n+1)\sigma_0^2$, which lead to the QMEC with $n$-order HG pointer being improved as:

$$S_{12}^{(n)} = (2n+1)^2 \qquad (14)$$

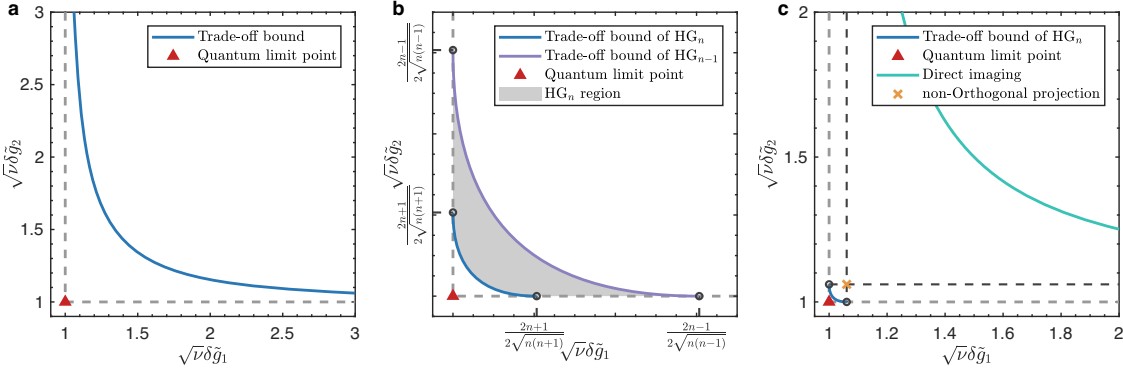

**Fig. 4 | Precision limits of the incompatible parameters.** The x-axis and y-axis are separately the normalized estimation errors of parameters $g_1$ and $g_2$. The gray dashed lines are separately the QCR bounds of $\tilde{g}_1$ and $\tilde{g}_2$. The cross point (red triangle in figure) of gray dashed lines is the QL point where both parameters achieve the ultimate precision. **a** Trade-off bound of Gaussian pointer, which is expressed by the blue solid curve, the region below this curve is forbidden by the inequality in Eq. (5). **b** Trade-off bound of HG pointer. The blue solid curve is the trade-off bound of $HG_n$ pointer, the purple solid curve is the trade-off bound of $HG_n$

–1 pointer. The gray shadow region between these two curves is named as $HG_n$ region, which is allowed for $HG_n$ pointer but forbidden for $HG_{n-1}$ pointer. **c** Practical precision limits of HG pointer when employing direct imaging and non-orthogonal projection measurements. The blue solid curve is the trade-off bound of $HG_n$ pointer. The green solid curve is the joint CCR bound under direct imaging measurement. The black dashed lines are the CCR bounds under with non-orthogonal projection measurement, where the cross point (yellow X mark) is the corresponding precision limit point.

Therefore, the corresponding trade-off bound of HG pointer can be calculated as:

$$2 - \frac{1}{\nu(\delta\tilde{g}_1)^2} - \frac{1}{\nu(\delta\tilde{g}_2)^2} + \frac{4\sqrt{n(n+1)}}{2n+1}$$
$$\times \sqrt{\left[1 - \frac{1}{\nu(\delta\tilde{g}_1)^2}\right]\left[1 - \frac{1}{\nu(\delta\tilde{g}_2)^2}\right]} \geq \frac{1}{(2n+1)^2} \quad (15)$$

where $\delta\tilde{g}_1 = |A_w|\sqrt{2n+1}\delta g_1/\sigma_0$ and $\delta\tilde{g}_2 = 2|A_w|\sigma_0\sqrt{2n+1}\delta g_2$. According to Eq. (12), the corresponding normalized parameter vector can be calculated as

$$\boldsymbol{\tilde{g}} = (\tilde{g}_1, \tilde{g}_2) = A_w\sqrt{2n+1}(g_1/\sigma_0, 2\sigma_0 g_2) \quad (16)$$

According to Eq. (15), the trade-off bound of $n$-order HG pointer is improved to a finite length curve with two endpoints:

$$\mathcal{P}_L^{(n)} = \sqrt{\nu}\left(\delta\tilde{g}_{1Q}^{min}, \delta\tilde{g}_{2L}^{min}\right) = \left(1, \frac{2n+1}{2\sqrt{n(n+1)}}\right)$$
$$\mathcal{P}_R^{(n)} = \sqrt{\nu}\left(\delta\tilde{g}_{1R}^{min}, \delta\tilde{g}_{2Q}^{min}\right) = \left(\frac{2n+1}{2\sqrt{n(n+1)}}, 1\right) \quad (17)$$

Though the QL point $\mathcal{P}_Q = (1, 1)$ of the incompatible parameters is practically unachievable with any classical measurement methods. However, we find that the trade-off bound of HG pointer is gradually closing to the QL point as the increasing of mode number $n$, because

$$\lim_{n\to\infty} \frac{2n+1}{2\sqrt{n(n+1)}} = \lim_{n\to\infty} \sqrt{1 + \frac{1}{4n(n+1)}} = 1 \quad (18)$$

From this relation, we can conclude that: when $n \to \infty$, the endpoints of trade-off bound $\mathcal{P}_L^{(n)} \to \mathcal{P}_Q$, $\mathcal{P}_R^{(n)} \to \mathcal{P}_Q$, which means that by employing high-order HG pointer, we can approach the quantum limits of the incompatible parameters simultaneously with some classical measurement methods in a practical system.

Here, we illustrated the trade-off bounds of $HG_n$ pointer and $HG_{n-1}$ pointer ($n > 1$) in Fig. 4b, where the blue solid curve is the trade-off bound of $HG_n$ pointer and the purple solid curve is the trade-off bound of $HG_{n-1}$ pointer. Moreover, the gray shadowed region between these

two curve is allowed for $HG_n$ pointer but forbidden for $HG_{n-1}$ pointer, which is named as $HG_n$ region.

The results illustrated in Fig. 4b show that the incompatible quantum limits for parameters $g_1$ and $g_2$ can be approached simultaneously by increasing the mode number of HG pointer. Moreover, it can be proved that this improvement depends on the uncertainty properties of the high-order HG state rather than its higher energy level. (See the Supplementary Note 2 for details.)

To implement the practical measurement of the incompatible parameters, we would like to investigate two widely used practical measurement strategies: direct imaging and non-orthogonal projection. Given a set of positive-operator-valued measure (POVM) $\hat{\Pi} = \{\hat{\Pi}_\lambda \mid \hat{\Pi}_\lambda \geq 0, \sum_\lambda \hat{\Pi}_\lambda = \hat{\mathbb{I}}\}$, the estimation covariance matrix of parameters $\boldsymbol{g}$ satisfies the classical Cramér-Rao (CCR) inequality: $\text{Cov}(\boldsymbol{g}, \hat{\Pi}) \geq \frac{1}{\nu}\mathcal{F}^{-1}$, where $\mathcal{F}$ is the classical Fisher information matrix (CFIM) with entry be calculated by[41]:

$$\mathcal{F}_{ij} = \sum_\lambda \frac{1}{\langle\hat{\Pi}_\lambda\rangle_{\boldsymbol{g}}} \frac{\partial\langle\hat{\Pi}_\lambda\rangle_{\boldsymbol{g}}}{\partial g_i} \frac{\partial\langle\hat{\Pi}_\lambda\rangle_{\boldsymbol{g}}}{\partial g_j} \quad (19)$$

where $\langle\hat{\Pi}_\lambda\rangle_{\boldsymbol{g}} = \langle\psi_{\boldsymbol{g}}|\hat{\Pi}_\lambda|\psi_{\boldsymbol{g}}\rangle$ is the measurement probability of state $|\psi_{\boldsymbol{g}}\rangle$ under operator $\hat{\Pi}_\lambda$. Substituting $|\psi_i\rangle = |n\rangle$ into $|\psi_f\rangle$, the final pointer state after the incompatible-parameters interaction can be calculated as:

$$|\psi_f\rangle \approx |n\rangle - \frac{1}{2}\left(\tilde{g}_1|\psi_{\hat{P}}\rangle + i\tilde{g}_2|\psi_{\hat{X}}\rangle\right) \quad (20)$$

where $|\psi_{\hat{P}}\rangle$ and $|\psi_{\hat{X}}\rangle$ are the generated states by operators $\hat{P}$ and $\hat{X}$:

$$|\psi_{\hat{P}}\rangle = \frac{1}{\sqrt{2n+1}}\left(\sqrt{n}|n-1\rangle - \sqrt{n+1}|n+1\rangle\right)$$
$$|\psi_{\hat{X}}\rangle = \frac{1}{\sqrt{2n+1}}\left(\sqrt{n}|n-1\rangle + \sqrt{n+1}|n+1\rangle\right) \quad (21)$$

For direct imaging method, the measurement operators can be denoted as $\hat{\Pi}^{DI} = \{|x\rangle\langle x| \mid x \in \mathbb{R}\}$. Substituting $\hat{\Pi}_{DI}$ into Eq. (19), the CFIM of direct imaging method can be obtained as:

$$\mathcal{F}_{DI}^{(n)} \approx \begin{pmatrix} (2n+1)\sigma_0^{-2}\text{Re}^2 A_w & 2\text{Re}A_w\text{Im}A_w \\ 2\text{Re}A_w\text{Im}A_w & 4(2n+1)\sigma_0^2\text{Im}^2 A_w \end{pmatrix} \quad (22)$$

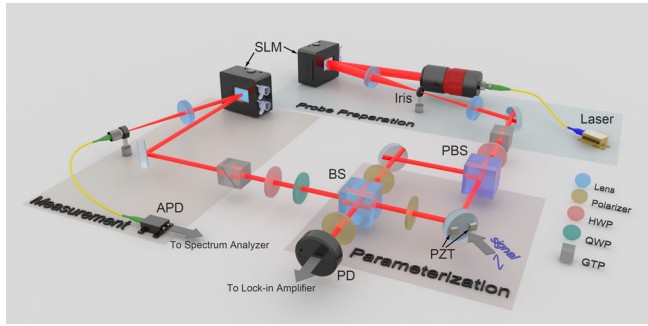

**Fig. 5 | Diagram of experimental setup.** The $n$-order HG beam is converted from an expanded Gaussian beam working at 780nm by an SLM and a spatial filter system. The pre-selection is implemented by a Glan-Taylor Polarizer and a HWP. And a polarized MZI is employed to implement the weak interaction procedure, where the incompatible parameters (position displacement and momentum kick) are introduced by a PZT-driven mirror. Then a quarter wave plate (QWP), a HWP, and another Glan-Taylor Polarizer are used to implement the post-selection. Finally, another SLM with a Fourier transfer lens is used to implement the non-orthogonal projection measurement, where the successfully projected photons are collected by an APD with an SMF.

Combining with the CCR inequalities, the normalized estimation errors of the incompatible parameters can be proved satisfying the inequality:

$$\frac{1}{\nu(\delta\tilde{g}_1)^2} + \frac{1}{\nu(\delta\tilde{g}_2)^2} \le \frac{4n(n+1)}{(2n+1)^2} \tag{23}$$

We illustrate this joint precision-limit bound in Fig. 4c with green solid curve, where the separate ultimate precisions of parameters are $\delta\tilde{g}_{1R}^{\min}$ and $\delta\tilde{g}_{2L}^{\min}$. Though $\delta\tilde{g}_{1R}^{\min}$ and $\delta\tilde{g}_{2L}^{\min}$ separately approach the quantum limits $\delta\tilde{g}_{1Q}^{\min}$ and $\delta\tilde{g}_{2Q}^{\min}$ as the increasing of mode number $n$, these two parameters can not approach their ultimate precision limits simultaneously based on the inequality in Eq. (23), which means that the direct imaging method is incapable to approach the QL point of incompatible parameters.

Drawing on the method of unambiguous quantum state discrimination[42,43], we devise a set of non-orthogonal projection operators $\hat{\Pi}_{\not\perp} = \{\hat{\Pi}_1 = |\psi_{\hat{X}}^\perp\rangle\langle\psi_{\hat{X}}^\perp|, \hat{\Pi}_2 = |\psi_{\hat{P}}^\perp\rangle\langle\psi_{\hat{P}}^\perp|, \hat{\mathbb{I}} - \hat{\Pi}_1 - \hat{\Pi}_2\}$, where the states:

$$\begin{aligned}|\psi_{\hat{X}}^\perp\rangle &= \frac{1}{\sqrt{2n+1}}\left(\sqrt{n+1}|n-1\rangle - \sqrt{n}|n+1\rangle\right) \\ |\psi_{\hat{P}}^\perp\rangle &= \frac{1}{\sqrt{2n+1}}\left(\sqrt{n+1}|n-1\rangle + \sqrt{n}|n+1\rangle\right)\end{aligned} \tag{24}$$

which are orthogonal to the parameters generated states $|\psi_{\hat{P}}\rangle$ and $|\psi_{\hat{X}}\rangle$. Therefore, it is easy to determine that the parameters $g_1$ and $g_2$ can be separately detected by the projection measurements $\hat{\Pi}_1$ and $\hat{\Pi}_2$.

Substituting the POVM $\hat{\Pi}_{\not\perp}$ into Eq. (24), the CFIM of non-orthogonal projection method can be obtained as:

$$\mathcal{F}_{\not\perp}^{(n)} \approx |A_w|^2 \begin{pmatrix} \frac{4n(n+1)}{(2n+1)\sigma_0^2} & 0 \\ 0 & \frac{16n(n+1)\sigma_0^2}{2n+1} \end{pmatrix} \tag{25}$$

which leads to the precision limits

$$\begin{cases} \delta\tilde{g}_1 & \ge \frac{2n+1}{2\sqrt{n(n+1)\nu}} \\ \delta\tilde{g}_2 & \ge \frac{2n+1}{2\sqrt{n(n+1)\nu}} \end{cases} \tag{26}$$

Combining with Eq. (18), it reveals that the QL point of the incompatible parameters can be approached asymptotically by employing

non-orthogonal projection measurement, and the corresponding practical precision limit in Eq. (26) is a point at:

$$\mathcal{P}_{\not\perp}^{(n)} = \sqrt{\nu}\left(\delta\tilde{g}_{1\not\perp}^{\min}, \delta\tilde{g}_{2\not\perp}^{\min}\right) = \frac{2n+1}{2\sqrt{n(n+1)}}\mathcal{P}_Q \tag{27}$$

Obviously, we can calculate that: when $n \to \infty$, the precision limit point $\mathcal{P}_{\not\perp}^{(n)} \to \mathcal{P}_Q$, which means that non-orthogonal projection measurement can approach the quantum limits of the incompatible parameters simultaneously.

## Experimental set-up and results

To experimentally verify that using HG pointer can approach the quantum limits of incompatible parameters simultaneously, we employ Hermite-Gaussian beam in an optical experiment, whose transverse spatial state can be expressed as $|u_n(z)\rangle = \hat{U}(z)|n\rangle$, where $\hat{U}(z) = \exp\left(\mathrm{i}\frac{1}{2k}\hat{P}^2 z\right)$ is the z-dependent propagation operator. Here $k = 2\pi/\lambda$ is the wave number of light beam and $z$ is the propagation distance begin from the beam waist. Therefore, the generators of the incompatible parameters $g_1$ and $g_2$ should evolve as $\hat{P}(z) = \hat{U}(z)\hat{P}\hat{U}^\dagger(z)$ and $\hat{X}(z) = \hat{U}(z)\hat{X}\hat{U}^\dagger(z)$.

In this experiment, we generate the $n$-order HG beam via a spatial light modulator (SLM) and a spatial filter system[44], as is shown in Fig. 5. The light beam from the laser working at 780nm is expanded for generating HG beams. Here, we choose the beam's polarization kets $|H\rangle$ and $|V\rangle$ as the basis of measured two-level system. The pre-selection state $|i\rangle = \frac{1}{\sqrt{2}}(|H\rangle + |V\rangle)$ is implemented by a Glan-Taylor polarizer (GTP) and a half wave plate (HWP). The pre-selected beam is injected to a polarized Mach-Zehnder interferometer (MZI), and a mirror driven by two piezoelectric transducer (PZT) chips is used to generate the tiny transverse spatial displacement $d$ and tiny angular tilt $\varphi$ (which leads to a tiny transverse momentum kick of $k\varphi$) simultaneously for the $|H\rangle$ component of light beam. Thus, the unitary evolution of this weak interaction procedure can be denoted as:

$$\hat{U}_w^{\exp} = \exp(-\mathrm{i}d\hat{P}\otimes\hat{A} - \mathrm{i}k\varphi\hat{X}\otimes\hat{A}) \tag{28}$$

where $\hat{A} = \frac{1}{2}\left(\hat{\mathbb{I}} + \hat{\sigma}_z\right)$, and $\hat{\sigma}_z = |H\rangle\langle H| - |V\rangle\langle V|$ is the Pauli operator.

After the weak interaction procedure in the polarized MZI, the light beam is post-selected by state $|f\rangle = \cos\left(\frac{\pi}{4} - \frac{\varepsilon}{2}\right)|H\rangle - \sin\left(\frac{\pi}{4} - \frac{\varepsilon}{2}\right)|V\rangle$, where $\varepsilon \ll 1$ (in our experiment, the post-selection angle is set as $\varepsilon = 5°$). Thus, the weak value can be calculated as

$$A_w = \frac{\langle f|\hat{A}|i\rangle}{\langle f|i\rangle} = \frac{1}{2}\left(\cot\frac{\varepsilon}{2} + 1\right) \approx \frac{1}{\varepsilon} \tag{29}$$

After the post-selection, another SLM is employed to implement the non-orthogonal projection measurement with a Fourier transfer lens and a spatial filtering from a single mode fiber (SMF) coupling detected photons to an avalanche photodiode (APD).

The optical length from the waist of HG beam to the signal mirror is denoted as $z_1$, the length from the signal mirror to the second SLM is denoted as $z_2$, and $z_1 + z_2 = z_0$. Thus, the unitary parameterization with incompatible parameters $g_1$ and $g_2$ is $\hat{U}(\boldsymbol{g}) = \hat{U}(z_2)\hat{U}_w^{\exp}\hat{U}^\dagger(z_2)$ in this experiment (see the Methods part), where $g_1$ and $g_2$ can be separately expressed as

$$g_1 = d + z_1\varphi, \quad g_2 = k\varphi \tag{30}$$

and the final beam state is then derived as (see the Methods part):

$$|u_f(z_0)\rangle \approx |u_n(z_0)\rangle - \frac{1}{2}\left(\tilde{g}_1|u_{\hat{P}}\rangle + \mathrm{i}\tilde{g}_2|u_{\hat{X}}\rangle\right) \tag{31}$$

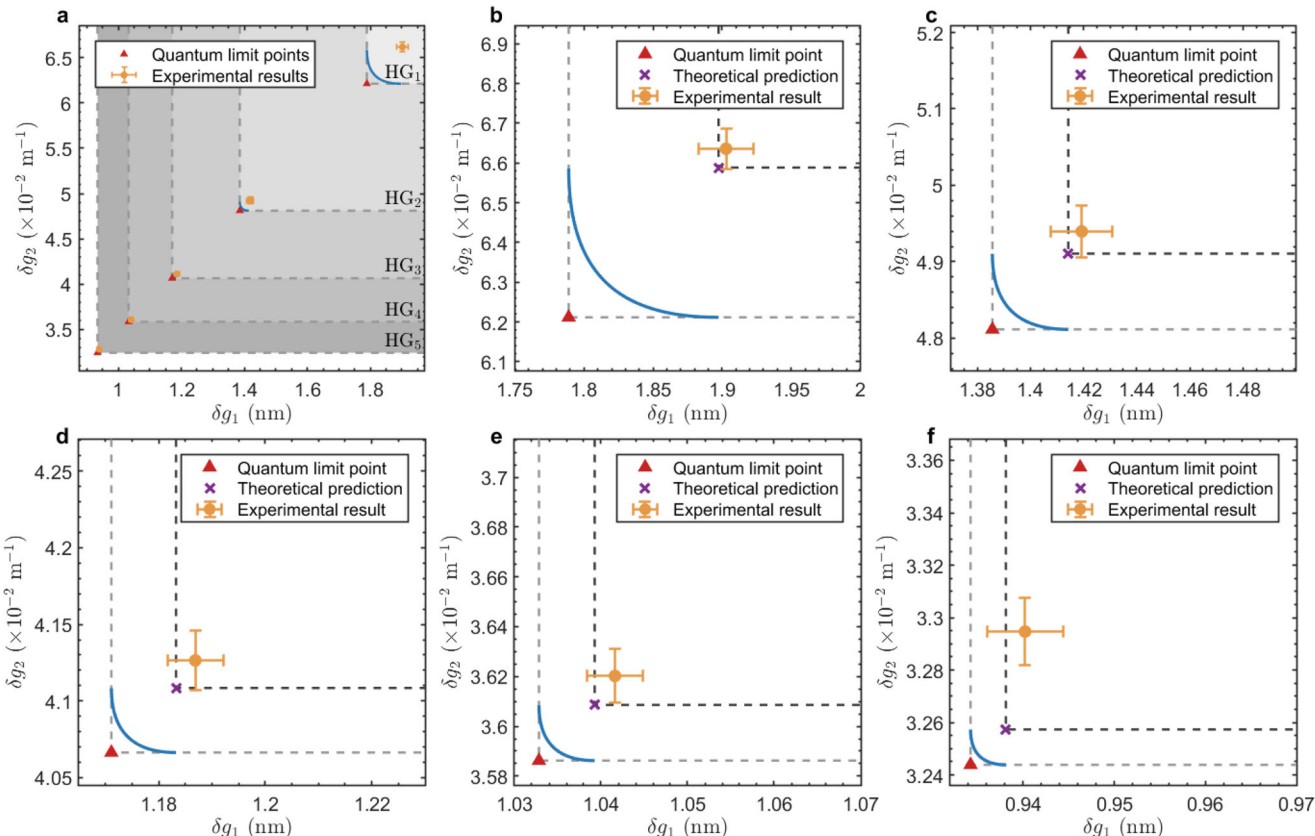

**Fig. 6 | Experimental results of minimum detectable parameters $g_1$ and $g_2$.** The experimental results are illustrated by the yellow points with error bar. The trade-off bounds of parameters $g_1$ and $g_2$ with different HG modes are represented by the blue solid curves. **a** Experimental results of $HG_1$ to $HG_5$ modes. The different HG regions are distinguished by the gray levels. The gray dashed lines are the QCR bounds of parameters $g_1$ and $g_2$ with different HG modes, and the cross points (red triangles) are the corresponding QL points. **b-f** Specific experimental results of $HG_1$ to $HG_5$ modes. The gray dashed lines are the QCR bounds of parameters $g_1$ and $g_2$, where the cross points (red triangles) are the theoretical QL points. The dark dashed lines are the experimental precision limits of parameters $g_1$ and $g_2$, where the cross points (purple cross marks) are the theoretical predictions of experimental results given by Eq. (36) and Eq. (37).

where $|u_{\hat{P}}\rangle = \hat{U}(z_0)|\psi_{\hat{P}}\rangle$ and $|u_{\hat{X}}\rangle = \hat{U}(z_0)|\psi_{\hat{X}}\rangle$ are the generated states by operators $\hat{P}(z)$ and $\hat{X}(z)$, $\tilde{g}_1 = A_w\sqrt{2n+1}g_1/\sigma_0$ and $\tilde{g}_2 = 2A_w\sqrt{2n+1}\sigma_0 g_2$ are still the normalized parameters regarding the parameters $g_1$ and $g_2$. Thus the non-orthogonal projection operators for HG beams are $\hat{\Pi}_1 = |u_{\hat{X}}^{\perp}\rangle\langle u_{\hat{X}}^{\perp}|$ and $\hat{\Pi}_2 = |u_{\hat{P}}^{\perp}\rangle\langle u_{\hat{P}}^{\perp}|$.

By displaying the non-orthogonal projection $\hat{\Pi}_1$ on the SLM, the parameter $g_1$ can be individually detected with projection probability

$$P_1 = \langle u_f(z_0)|\hat{\Pi}_1|u_f(z_0)\rangle = \frac{n(n+1)}{(2n+1)^2}\tilde{g}_1^2 \qquad (32)$$

Then displaying the non-orthogonal projection $\hat{\Pi}_2$ on the SLM, the parameter $g_2$ can be also individually detected with projection probability

$$P_2 = \langle u_f(z_0)|\hat{\Pi}_2|u_f(z_0)\rangle = \frac{n(n+1)}{(2n+1)^2}\tilde{g}_2^2 \qquad (33)$$

In this experiment, we generate the position displacement and angular tilt signals simultaneously for light beam via a PZT driven mirror, where two asynchronous cosine driving signals with frequency $f = 2\,\text{kHz}$ and relative phase $\theta = 4°$ are exerted on the two PZT chips separately. Besides the 2 kHz driving signals, the initial displacement and tilt bias errors of the mirror can not be negligible in practice, which leads to the initial biases $g_1^{\Delta}$ and $g_2^{\Delta}$ for the parameters $g_1$ and $g_2$. Thus, the total signals of the incompatible parameters is denoted as

$g_1^{\text{tot}} = g_1^{\Delta} + g_1\cos(2\pi f t)$ and $g_2^{\text{tot}} = g_2^{\Delta} + g_2\cos(2\pi f t)$ separately. It is easy to determine that $g_1 \ll g_1^{\Delta} \ll 1$ and $g_1 \ll g_1^{\Delta} \ll 1$ (see the Methods part).

In practice, before exerting the driving signals, we projected the final beam state on $|u_n(z_0)\rangle$ to fix the effective sample number $\nu$ for different HG modes, which is obtained by the detected optical power $I_0$ (experimental values of $I_0$ and $\nu$ are given in the Method part). Then exert the driving signals on PZT chips and project the final beam on state $|u_{\hat{X}}^{\perp}\rangle$ and $|u_{\hat{P}}^{\perp}\rangle$ separately, whose corresponding detected optical powers are denoted as $I_1$ and $I_2$. Finally, the detected power signals were inputted into the spectrum analyzer, where we can independently demodulate the parameters $g_1$ and $g_2$ from the corresponding peak powers at $f = 2\,\text{kHz}$:

$$
\begin{aligned}
I_1^{(2\,\text{kHz})} &= \frac{2n(n+1)}{(2n+1)\sigma_0^2\varepsilon^2}g_1^{\Delta}g_1 I_0 \\
I_2^{(2\,\text{kHz})} &= \frac{8n(n+1)\sigma_0^2}{(2n+1)\varepsilon^2}g_2^{\Delta}g_2 I_0
\end{aligned}
\qquad (34)
$$

Here we only concern with the shot noise in experiment, then the theoretical detected peak signal-to-noise ratios (SNR) with two non-orthogonal projections can be obtained as:

$$
\begin{aligned}
\text{SNR}_1 &= \frac{I_1^{(2\,\text{kHz})}}{\delta I_1} = \sqrt{\frac{n(n+1)\nu}{(2n+1)\varepsilon^2}}\frac{2g_1}{\sigma_0} \\
\text{SNR}_2 &= \frac{I_2^{(2\,\text{kHz})}}{\delta I_2} = \sqrt{\frac{n(n+1)\nu}{(2n+1)\varepsilon^2}}4\sigma_0 g_2
\end{aligned}
\qquad (35)
$$

**Table 1 | Experimental results**

| HG modes | SNR$_1$ = 1[c] | Driving Voltages[a] | | $\delta \boldsymbol{g}_{1\,\text{det}}^{\min}$ | $\delta \boldsymbol{g}_{2\,\text{det}}^{\min}$ | $\delta \boldsymbol{d}_{\text{det}}^{\min}$ | Parameters[b] |
| | | SNR$_2$ = 1[d] | | | | | $\delta \boldsymbol{\varphi}_{\text{det}}^{\min}$ |
|---|---|---|---|---|---|---|---|
| HG$_1$ | 73.10 mV | 107.03 mV | | 1.90 nm | $6.62 \times 10^{-2}$ m$^{-1}$ | 2.94 nm | 8.22 nrad |
| HG$_2$ | 54.53 mV | 79.67 mV | | 1.42 nm | $4.93 \times 10^{-2}$ m$^{-1}$ | 2.19 nm | 6.12 nrad |
| HG$_3$ | 45.59 mV | 66.56 mV | | 1.19 nm | $4.12 \times 10^{-2}$ m$^{-1}$ | 1.83 nm | 5.11 nrad |
| HG$_4$ | 40.01 mV | 58.39 mV | | 1.04 nm | $3.62 \times 10^{-2}$ m$^{-1}$ | 1.60 nm | 4.48 nrad |
| HG$_5$ | 36.12 mV | 53.14 mV | | 0.94 nm | $3.29 \times 10^{-2}$ m$^{-1}$ | 1.45 nm | 4.08 nrad |

[a]Driving voltages (peak-to-peak value) of PZT chips when SNR$_1$ = 1 and SNR$_2$ = 1 with different HG modes.
[b]These columns are the experimental results of minimum detected values of parameters $g_1$, $g_2$, and $g_1$.
[c]SNR$_1$ corresponds to the $\hat{\Pi}_1$ projection measurement, where parameter $g_1$ can be directly demodulated.
[d]SNR$_2$ corresponds to the $\hat{\Pi}_2$ projection measurement, where parameter $g_2$ can be directly demodulated.

When SNR$_1$ = 1, the minimum detectable parameter $g_1$ is:

$$\delta g_{1\,\text{det}}^{\min} = \sqrt{\frac{(2n+1)\varepsilon^2}{n(n+1)\nu}} \frac{\sigma_0}{2} \tag{36}$$

Similarly, when SNR$_2$ = 1, the minimum detectable parameter $g_2$ is:

$$\delta g_{2\,\text{det}}^{\min} = \sqrt{\frac{(2n+1)\varepsilon^2}{n(n+1)\nu}} \frac{1}{4\sigma_0} \tag{37}$$

Experimentally, the spatial displacement $d$ and angular tilt $\varphi$ of light beam can be estimated from the detected parameters $g_1$ and $g_2$ as $d = g_1 - \frac{z_1}{k} g_2$ and $\varphi = g_2/k$, which leads to the corresponding estimating errors be calculated as

$$\delta d = \sqrt{(\delta g_1)^2 + \frac{z_1^2}{k^2}(\delta g_2)^2}, \quad \delta\varphi = \frac{\delta g_2}{k} \tag{38}$$

Thus, the theoretical minimum detectable spatial displacement and angular tilt of light beam in our experiment can be obtained as:

$$\delta d_{\text{det}}^{\min} = \sqrt{\left(1 + \frac{z_1^2}{b^2}\right)\frac{(2n+1)\varepsilon^2}{n(n+1)\nu}} \frac{\sigma_0}{2}$$
$$\delta\varphi_{\text{det}}^{\min} = \sqrt{\frac{(2n+1)\varepsilon^2}{n(n+1)\nu}} \frac{\sigma_0}{2b} \tag{39}$$

where $b = 2k\sigma_0^2$ is the Rayleigh range.

In our experimental scheme, parameters $g_1$ and $g_2$ are able to directly detected by applying non-orthogonal projection measurement. Thus we illustrate the experimental precisions of parameters $g_1$ and $g_2$ in Fig. 6. The experimental results of minimum detectable parameters $g_1$ and $g_2$ are plotted without normalized, therefore the QL point of each HG mode is difference, and the HG regions turn into the 'L'-type. As is shown in Fig. 6a, our experimental precision points of different HG modes are exactly in the corresponding HG regions. In Fig. 6b to Fig. 6f, we illustrate the experimental results of HG$_1$ mode to HG$_5$ mode separately. In these sub-figures, we also plot the theoretical prediction of experimental precision for every HG mode, which is derived directly from Eq. (36) and Eq. (37) with the fixed experimental photons number $\nu = 1.05 \times 10^7$, post-selection angle $\varepsilon = 5°$ and beam waist radius $w_0 = 2\sigma_0 = 240$ μm.

To directly exhibit our experimental results, we also list the driving voltages of PZT chips when SNR$_1$ = 1 and SNR$_2$ = 1, and the practical minimum detected values of parameters $g_1$, $g_2$, and the corresponding spatial displacement $d$ and angular tilt $\varphi$ with different HG modes in Table 1. As results, we finally achieve the 1.45 nm precision on the light beam's spatial displacement and the 4.08 nrad precision on the light beam's angular tilt when measuring them simultaneously with 5-order HG beam.

## Geometrical properties of QMEC

In the quantum metrological process, the parameterization evolution $\hat{U}(\boldsymbol{g})$ projects the probe state $|\psi\rangle$ in Hilbert space to the parameterized state $|\psi_{\boldsymbol{g}}\rangle$ in parameter space. Theoretically, the matrix elements of $\mathcal{Q}$ are identical to the Fubini-Study metric[45] for pure state, which is a second-order tensor in the projective Hilbert space (parameter space). Thus, the QFIM can be naturally extended to the quantum geometric tensor (QGT)[46,47], whose entry is obtained by:

$$\mathcal{T}_{ij} = \frac{\partial\langle\psi_{\boldsymbol{g}}|}{\partial g_i}\left(1 - |\psi_{\boldsymbol{g}}\rangle\langle\psi_{\boldsymbol{g}}|\right)\frac{\partial|\psi_{\boldsymbol{g}}\rangle}{\partial g_j} \tag{40}$$

Obviously, QGT is a complex metric in the projective Hilbert space with properties:

$$\text{Re}\,\mathcal{T} = \frac{\mathcal{Q}}{4}, \quad \text{Im}\,\mathcal{T} = -\frac{\mathcal{C}}{2} \tag{41}$$

where $\mathcal{C}$ is the Berry curvature (BC) defined on the parameter space. For quantum estimation theory, the non-diagonal elements of BC indicate the non-commutativity of different parameters, which means the larger the BC is, the harder to estimate two different parameters simultaneously. In Lu and Wang's work[27], they related the trade-off bound to the quantum geometric tensor. Then the QMEC can be calculated as (see the Supplementary Note 1 for derivation):

$$\mathcal{S}_{ij} = \frac{\mathcal{Q}_{ii}\mathcal{Q}_{jj}}{4\mathcal{C}_{ij}^2} \tag{42}$$

where the geometric properties on parameter space are involved. Especially, the QMEC indicates a normalized curvature on the parameter space:

$$\tilde{\mathcal{C}}_{ij} = \frac{2\mathcal{C}_{ij}}{\sqrt{\mathcal{Q}_{ii}\mathcal{Q}_{jj}}} \tag{43}$$

where $\tilde{\mathcal{C}}_{ij}^2 = 1/\mathcal{S}_{ij}$. Traditionally, the Berry curvature $\mathcal{C}_{ij}$, i.e., the weak commutative condition is concerned alone in evaluating the incompatibility of different parameters on multiparameter estimation in Eq. (2). However, the geometrical properties of the QMEC in our work reveals that the Berry curvature alone is insufficient in evaluating the incompatibility, the metric property on the parameter space, i.e., the QFIM influences the attainability of simultaneous quantum limits on multiparameter estimation. Thus, the normalized curvature $\tilde{\mathcal{C}}_{ij}$ is more appropriate to describe the curvature property on the parameter space.

## Comparison of trade-off bound and Holevo bound

The Holevo CR bound is another wide studied lower bound of estimating variances on multiparameter quantum estimation, which

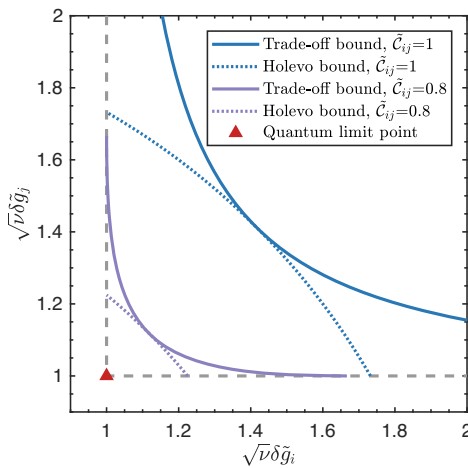

**Fig. 7 | Comparison of the trade-off bound and the Holevo CR bound for two incompatible parameters $g_i$ and $g_j$.** The gray dashed lines are the QCR bounds, where the cross point (red triangle) is the corresponding QL point. The blue solid curve stands for the trade-off bound with normalized Berry curvature $\bar{\mathcal{C}}_{ij} = 1$, and the blue dotted curve stands for the Holevo bound with normalized Berry curvature $\bar{\mathcal{C}}_{ij} = 1$. The purple solid curve stands for the trade-off bound with normalized Berry curvature $\bar{\mathcal{C}}_{ij} = 0.8$, and the purple dotted curve stands for the Holevo bound with normalized Berry curvature $\bar{\mathcal{C}}_{ij} = 0.8$.

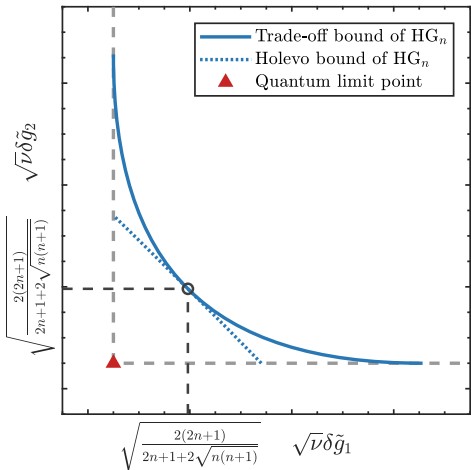

**Fig. 8 | Comparison of the trade-off bound and the Holevo CR bound for parameters $g_1$ and $g_2$ in our post-selected measurement scenario with $HG_n$ pointer.** The gray dashed lines are the QCR bounds, where the cross point (red triangle) is the corresponding QL point. The blue solid curve stands for the trade-off bound given by Eq. (15), and the blue dotted curve is the corresponding Holevo bound with $HG_n$ pointer.

reveals the quantum limits on weighted mean errors and is proven attainable within collective measurements[48,49]. However, the Holevo CR bound is difficult to completely identify the trade-off curve regarding the attainable precision limits on estimating different parameters. In this part, we would like to give the Holevo CR bound on estimating two incompatible parameters of a parameterized pure state, and compare it with our trade-off bound.

Theoretically, the analytic expression of Holevo bound is difficult to be obtained for the generalized multiparameter estimation scenario, and numerical algorithms are widely employed for solving it[50,51]. In our work, we concern with the measurement scenario of two incompatible parameters for a pure state model, where the corresponding Holevo bound is given as:

$$\frac{\nu(\delta\tilde{g}_i)^2 + \nu(\delta\tilde{g}_j)^2}{2} \geq \frac{2}{1 + \sqrt{1 - \bar{\mathcal{C}}_{ij}^2}} \tag{44}$$

which is first derived by Matsumoto at[52]. $\delta\tilde{g}_i$ and $\delta\tilde{g}_j$ are the normalized estimation errors for incompatible parameters $g_i$ and $g_j$, $\bar{\mathcal{C}}_{ij}$ is the normalized Berry curvature defined in Eq. (43). By comparing our trade-off bound of Eq. (5) with the Holevo bound in Fig. 7, we can see that the Holevo bound is exactly tangent to the trade-off bound at $\delta\tilde{g}_i = \delta\tilde{g}_j$. From the results in Fig. 7, we can conclude that though the Holevo bound gives a tight bound of weighted mean errors for two incompatible parameters, it is less informative than the trade-off bound employed in our work.

Furthermore, we concern with the Holevo bound in our post-selected measurement scenario with $HG_n$ pointer, where the normalized Berry curvature is given as $\bar{\mathcal{C}}_{12} = \frac{1}{2n+1}$. Then the Holevo bound for the incompatible parameters $g_1$ and $g_2$ generated by the momentum operator $\hat{P}$ and position operator $\hat{X}$ can be calculated as

$$\frac{\nu(\delta\tilde{g}_1)^2 + \nu(\delta\tilde{g}_2)^2}{2} \geq \frac{2(2n+1)}{2n+1+2\sqrt{n(n+1)}} \tag{45}$$

By comparing it with the trade-off bound of Eq. (15) in Fig. 8, we can see that the Holevo bound is tangent to the trade-off bound at

$\sqrt{\nu}\delta\tilde{g}_i = \sqrt{\nu}\delta\tilde{g}_j = \sqrt{\frac{2(2n+1)}{2n+1+2\sqrt{n(n+1)}}}$, and the trade-off bound is more informative than the Holevo bound.

In summary, this study has identified a practical way to approach the QL point of incompatible parameters asymptotically, which was considered unachievable. By connecting the estimation errors of parameters with the variances of corresponding generators, we have proposed a criterion to improve the trade-off precision bound from the physical insight. This significant criterion indicates that the incompatibility is able to be mitigated via increasing the corresponding generators' variances simultaneously, where devising an appropriate probe state is important. For demonstration, we have build up a practical scheme for measuring the incompatible parameters of momentum and position simultaneously in a quantum system, and a corresponding optical experiment has also been implemented on the simultaneous measurement of light beam's transverse displacement and angular tilt. By employing the HG states as the probe, the QL point of the incompatible parameters has been approached asymptotically in both theoretical frame and experimental results. Furthermore, our method is able to improve the measurement precisions, because increasing the generators' variances also decreases the ultimate estimation errors of unknown parameters. Experimentally, we have achieved the precisions up to 1.45 nm and 4.08 nrad on the simultaneous measurement of spatial displacement and angular tilt of light. Although only the spatial incompatible parameters are experimentally exemplified, our theoretical analysis is perfectly applicable to the homodyne detection of coherent light in the phase space[53]. Therefore, the theoretical and experimental findings in this study not only contribute to develop and refine the quantum multiparameter estimation theory, but also have potential for other quantum optics applications, such as quantum communications[54,55] and superresolution[8,56].

## Methods

### Technical advantages of weak value
To practically demonstrate our theory, we employed the post-selected weak measurement scheme with weak value amplification technique to measure the incompatible parameters simultaneously. The weak value $A_w \approx 1/\varepsilon$ takes no enhancement for the theoretical minimum detectable parameters because the detected photons number

$\nu = |\langle f|i\rangle|^2 \nu_0 = \sin^2\frac{\varepsilon}{2}\nu_0$ is attenuated by the post-selection, where $\nu_0$ is the photons number before post-selection. However, the weak value amplification technology has been proved efficient for suppressing technical noises, such as reflection of optical elements[2] and detector saturation[33,57,58]. Especially, the detector saturation is non-negligible in our experiment for the saturation power of our APD detector is only 1.54 nW. Considering the projection demodulation of SLM, only 10% photons can be modulated on the 1st-order diffraction, so the maximum received power of our detector is -154 pW, which is easily saturated without post-selection. For example, the efficient detected light power in our experiment is $I_0 = 49.09$ pW, and the post-selection angle $\varepsilon = 5°$. Therefore, for the post-selection-free scheme, a $I_0/\sin^2\frac{\varepsilon}{2} = 25.8$ nW detected light power is needed to achieve the same precision of the post-selected scheme, which is far lager than the saturation power of the APD detector.

## Experimental materials

The laser employed in this experiment is a Distributed Bragg Reflector (DBR) Single-Frequency Laser of of Thorlabs Inc. (part number: DBR780PN), which works at $\lambda = 780$ nm with 1 MHz typical linewidth. To generate the high-order HG beams, we used a SLM of Hamamatsu Photonics (part number: X13138-02), which has $1272 \times 1024$ pixels with 12.5 μm pixel pitch. The focal length of the Fourier lens in the 4-f system is 5 cm. A 200 μm square pinhole is used as the spatial filter.

In this work, we set up a polarized MZI to generate the position displacement and momentum kick interaction simultaneously for the weak measurement scheme. To improve th beam's degree of polarization, two additional polarizers were inserted into two optical arms of the interferometer. Moreover, a lock-in amplifier was used to stabilize the relative phase of two optical path in the MZI. In the experiment, we pasted 2 PZTs on the back of the signal mirror (see the Experimental materials part). Then we exerted two $f = 2$ kHz cosine driving signals with a relative phase $\theta$ on the two PZTs separately. The interval between these two PZTs is 20 mm, and the part number of these PZTs is NAC2013 of Core Tomorrow Company, which shifts 22nm with 1V driving voltage. Thus, when setting the relative phase $\theta = 0°$, there is only a 11 nm shift signal of mirror with 1 $V_{pp}$ driving Voltage, which leads to a $d = 15.56$ nm transverse displacement signal of light beam. In contrast, when setting the relative phase $\theta = 180°$, there is only a 1.1 μrad tilt signal of mirror ($\varphi = 2.2$ μrad angular tilt of light beam) with 1 $V_{pp}$ driving Voltage, which leads to a $k\varphi = 17.72$ m$^{-1}$ momentum kick signal of light beam, where $k = 2\pi/\lambda = 8.06 \times 10^6$ m$^{-1}$ is the wave number of laser beam.

In this experiment, we modulated the waist radius of fundamental Gaussian beam as $w_0 = 2\sigma_0 = 240$ μm and set the relative phase of driving signal as $\theta = 4°$. Then the 1 $V_{pp}$ driving signal of PZTs corresponds to the unit spatial displacement amplitude $d^u \approx 15.55$ nm/V and the unit angular tilt amplitude $\varphi^u \approx 76.78$ nrad/V. Besides, the misaligned bias of PZT mirror caused an initial displacement bias $d^\Delta$, which is on the $10^4$ nm scale (scale of $w_0/10$), and an initial angular tilt bias $\varphi^\Delta$, which is on the mrad scale. Thus, the total displacement signal is $d^{tot} = d^\Delta + d\cos(2\pi f t)$ and the total tilt signal $\varphi^{tot} = \varphi^\Delta + \varphi\sin(2\pi f t)$. It is easy to determine that $d \ll d^\Delta \ll 1$ and $\varphi \ll \varphi^\Delta \ll 1$. Moreover, the optical lengths $z_1 = -27.2$ cm, $z_2 = 64$ cm in our experiment. Thus, the total modulated signals of the incompatible parameters $g_1^{tot} = d^{tot} + z_1\varphi^{tot} = g_1^\Delta + g_1\cos(2\pi f t)$ and $g_2^{tot} = k\varphi^{tot} = g_2^\Delta + g_2\cos(2\pi f t)$. Thus, the modulated signals' amplitudes of the incompatible parameters at $f = 2$ kHz are $g_1 = \sqrt{d^2 + z_1^2\varphi^2}$ and $g_2 = k\varphi$, which lead to $g_1^u \approx 26.03$ nm/V and $g_2^u \approx 0.62$ m$^{-1}$/V.

In practice, by projecting the final beam state on $|u_n(z_0)\rangle$, we determined the effective sample number $\nu = N\tau$, where $N$ is the photon number in unit time and $\tau$ is the detecting time length. In this experiment, a Si Avalanche Photodetector (part number: APD440A of Thorlabs Inc.) was employed to receive the projection signal, which has

maximum conversion gain of $2.65 \times 10^9$ V/W and 100 kHz bandwidth. The detected optical power of APD on state $|u_n(z_0)\rangle$ was fixed as $I_0 = 49.06$ pW for different HG modes, which leads to $N = 1.92 \times 10^8$ s$^{-1}$. Then the detected voltage signal was analyzed by the spectrum analyzer module of Moku:Lab, which is a reconfigurable hardware platform produced by Liquid instruments. The resolution bandwidth (RWB) of our spectrum analyzer was 18.34 Hz, which leads to the detecting time of $\tau = 54.53$ ms. Thus, the effective sample number in our experiment is fixed as $\nu = 1.05 \times 10^7$.

## Operator algebra for HG state

From the view of quantum mechanics, $\psi_n(x)$ is the time-independent solution for the Schrödinger equation of harmonic oscillators:

$$i\frac{\partial\psi}{\partial t} = \left(\sigma_0^2\hat{P}^2 + \frac{1}{4\sigma_0^2}\hat{X}^2\right)\psi \tag{46}$$

For eigenvalue $E_n = (n + \frac{1}{2})$, the corresponding eigenket can be obtained as:

$$|n\rangle = \int dx\,\psi_n(x)|x\rangle \tag{47}$$

Thus, we can employ the mode creation (annihilation) operators $\hat{a}(\hat{a}^\dagger)$ for HG state:

$$\begin{cases} \hat{a}|n\rangle = \sqrt{n}|n-1\rangle \\ \hat{a}^\dagger|n\rangle = \sqrt{n+1}|n+1\rangle \end{cases} \tag{48}$$

Then the higher-order HG state can be obtained from the fundamental Gaussian state with the creation operator:

$$|n\rangle = \frac{1}{\sqrt{n!}}(\hat{a}^\dagger)^n|0\rangle \tag{49}$$

Besides, the momentum operator $\hat{P}$ and position operator $\hat{X}$ can be also represented by creation and annihilation operators:

$$\begin{aligned} \hat{P} &= \frac{1}{i2\sigma_0}(\hat{a} - \hat{a}^\dagger) \\ \hat{X} &= \sigma_0(\hat{a} + \hat{a}^\dagger) \end{aligned} \tag{50}$$

Substituting Eq. (50) into Eq. (7), it is easy to calculate the final pointer state with $n$-order HG state as:

$$|\psi_f\rangle \approx |n\rangle - A_w\sqrt{2n+1}\left(\frac{g_1}{2\sigma_0}|\psi_{\hat{P}}\rangle + i\sigma_0 g_2|\psi_{\hat{X}}\rangle\right) \tag{51}$$

where the generated states $|\psi_{\hat{P}}\rangle$ and $|\psi_{\hat{X}}\rangle$ have been given in Eq. (21).

## Propagation of HG beams

The z-dependent wave function of $n$-order HG beam is[40]

$$u_n(x,z) = \sqrt{\frac{\sigma_0}{\sigma(z)}}\psi_n\left[\frac{\sigma_0 x}{\sigma(z)}\right]\exp\left[\frac{ikx^2}{2q(z)} - i(n + \frac{1}{2})\chi(z)\right] \tag{52}$$

where the three z-dependent parameters: spatial variance of fundamental Gaussian beam $\sigma^2$, Gouy phase $\chi$ and curvature radius of the wavefront $q$ can be determined by equalities:

$$\frac{1}{2\sigma^2(z)} - \frac{ik}{q(z)} = \frac{k}{b + iz}, \quad \tan\chi(z) = \frac{z}{b} \tag{53}$$

Here $k = 2\pi/\lambda$ is the wave number of light beam and $b = 2k\sigma_0^2$ is the Rayleigh range[59]. Then we denote $|u_n(z)\rangle = \int dx\,u_n(x,z)|x\rangle$,

obviously, $|u_n(0)\rangle = |n\rangle$. The HG beams are solutions of the paraxial wave equation

$$\frac{\partial^2}{\partial x^2} u_n(x,z) = -\mathrm{i}2k\frac{\partial}{\partial z}u_n(x,z)$$

which can be rewritten as

$$\frac{\mathrm{d}}{\mathrm{d}z}|u_n(z)\rangle = -\frac{\mathrm{i}}{2k}\hat{P}^2|u_n(z)\rangle \tag{54}$$

This equation has the formal solution $|u_n(z)\rangle = \hat{U}(z)|u_n(0)\rangle$ with the propagation operator

$$\hat{U}(z) = \exp\left(-\frac{\mathrm{i}}{2k}\hat{P}^2 z\right) \tag{55}$$

Thus, the z-dependent creation (annihilation) operators are given by $\hat{a}(z) = \hat{U}(z)\hat{a}\hat{U}^\dagger(z)$ and $\hat{a}^\dagger(z) = \hat{U}(z)\hat{a}^\dagger\hat{U}^\dagger(z)$. Hence, the higher-order HG beam state can also be obtained by the fundamental state according to

$$|u_n(z)\rangle = \frac{1}{\sqrt{n!}}\left[\hat{a}^\dagger(z)\right]^n|u_0(z)\rangle \tag{56}$$

Moreover, the z-dependent momentum and position operators $\hat{P}(z)$ and $\hat{X}(z)$ for free propagation can be derived as:

$$\hat{P}(z) = \hat{U}(z)\hat{P}\hat{U}^\dagger(z) = \hat{P}$$
$$\hat{X}(z) = \hat{U}(z)\hat{X}\hat{U}^\dagger(z) = X - \frac{z}{k}\hat{P} \tag{57}$$

According to the experimental set-up, the final state of the whole system can be calculated by:

$$\begin{aligned}|\Psi_f\rangle &= |f\rangle\langle f|\hat{U}(z_2)\hat{U}_{\mathrm{w}}^{\mathrm{exp}}\hat{U}(z_1)|u_n(0)\rangle|i\rangle \\ &= |f\rangle\langle f|\hat{U}(z_2)\hat{U}_{\mathrm{w}}^{\mathrm{exp}}\hat{U}^\dagger(z_2)|u_n(z_0)\rangle|i\rangle \\ &= |f\rangle\langle f|\hat{U}(\boldsymbol{g})|u_n(z_0)\rangle|i\rangle\end{aligned} \tag{58}$$

where $\hat{U}(\boldsymbol{g}) = \hat{U}(z_2)\hat{U}_{\mathrm{w}}^{\mathrm{exp}}\hat{U}^\dagger(z_2)$ is the unitary parameterization in the experiment, and it can be expressed with $\hat{P}(z_0)$ and $\hat{X}(z_0)$ by:

$$\begin{aligned}\hat{U}(\boldsymbol{g}) &= \hat{U}(z_2)\hat{U}_{\mathrm{w}}^{\mathrm{exp}}\hat{U}^\dagger(z_2) \\ &= \exp\left[-\mathrm{i}d\hat{P}(z_2)\otimes\hat{A} - \mathrm{i}k\varphi\hat{X}(z_2)\otimes\hat{A}\right] \\ &= \exp\left[-\mathrm{i}g_1\hat{P}(z_0)\otimes\hat{A} - \mathrm{i}g_2\hat{X}(z_0)\otimes\hat{A}\right]\end{aligned} \tag{59}$$

where $g_1 = d + z_1\varphi$ and $g_2 = k\varphi$. This expression is derived from the relations:

$$\hat{P}(z_0) = \hat{P}(z_2) = \hat{P}$$
$$\hat{X}(z_0) - \hat{X}(z_2) = -\frac{z_0 - z_2}{k}\hat{P} = -\frac{z_1}{k}\hat{P}(z_0) \tag{60}$$

Substituting Eq. (59) into Eq. (58), the final beam state $|u_f(z_0)\rangle$ in Eq. (31) is finally calculated by

$$\begin{aligned}|u_f(z_0)\rangle &\approx \left[1 - \mathrm{i}A_w g_1\hat{P}(z_0) - \mathrm{i}A_w g_2\hat{X}(z_0)\right]|u_n(z_0)\rangle \\ &= \hat{U}(z_0)|\psi_f\rangle\end{aligned} \tag{61}$$

Thus, the projective states $|u_{\hat{P}}\rangle = \hat{U}(z_0)|\psi_{\hat{P}}\rangle$ and $|u_{\hat{X}}\rangle = \hat{U}(z_0)|\psi_{\hat{X}}\rangle$ in the experiment. Moreover, the normalized parameters $\tilde{g}_1$ and $\tilde{g}_2$ in

the experiment are still given by $\tilde{g}_1 = A_w\sqrt{2n+1}g_1/\sigma_0$ and $\tilde{g}_2 = 2A_w\sigma_0\sqrt{2n+1}g_1$.

## SNR calculation

In the experiment, the detected optical power of APD on state $|u_n(z_0)\rangle$ is denoted as $I_0$, which was fixed for different HG modes. Exerting the driving signals on PZT chips and separately projecting the final beam on states $|u_{\hat{X}}^\perp\rangle$ and $|u_{\hat{P}}^\perp\rangle$, then the corresponding detected optical powers are calculated as:

$$\begin{aligned}I_1 = P_1 I_0 &= \frac{n(n+1)}{(2n+1)^2}(\tilde{g}_1^{\mathrm{tot}})^2 I_0 \\ &\approx \frac{n(n+1)}{(2n+1)\sigma_0^2\varepsilon^2}(g_1^\Delta)^2 I_0 \\ &+ \frac{2n(n+1)}{(2n+1)\sigma_0^2\varepsilon^2}g_1^\Delta g_1\cos(2\pi ft)I_0\end{aligned} \tag{62}$$

$$\begin{aligned}I_2 = P_2 I_0 &= \frac{n(n+1)}{(2n+1)^2}(\tilde{g}_2^{\mathrm{tot}})^2 I_0 \\ &\approx \frac{4n(n+1)\sigma_0^2}{(2n+1)\varepsilon^2}(g_2^\Delta)^2 I_0 \\ &+ \frac{8n(n+1)\sigma_0^2}{(2n+1)\varepsilon^2}g_2^\Delta g_2\cos(2\pi ft)I_0\end{aligned} \tag{63}$$

Inputting the detected power signals of APD into the spectrum analyzer, where we can independently demodulate the parameters $g_1$ and $g_2$ at $f = 2\,\mathrm{kHz}$ from the corresponding peak powers $I_1^{(2\,\mathrm{kHz})}$ and $I_2^{(2\,\mathrm{kHz})}$ in Eq. (34). Besides, the corresponding detected shot noises of APD when displaying measurements $\hat{\Pi}_1 = |u_{\hat{X}}^\perp\rangle\langle u_{\hat{X}}^\perp|$ and $\hat{\Pi}_2 = |u_{\hat{P}}^\perp\rangle\langle u_{\hat{P}}^\perp|$ can be separately calculated as:

$$\begin{aligned}\delta I_1 &\approx \sqrt{\frac{n(n+1)}{(2n+1)\varepsilon^2}}\frac{g_1^\Delta}{\sigma_0}\delta I_0 \\ \delta I_2 &\approx \sqrt{\frac{n(n+1)}{(2n+1)\varepsilon^2}}2\sigma_0 g_2^\Delta\delta I_0\end{aligned} \tag{64}$$

Theoretically, the effective optical power of final beam state is $I_0 = \gamma\nu/\tau$, where $\gamma$ is the single photon's energy at $\lambda = 780\,\mathrm{nm}$. Then the corresponding shot noise of final beam state is given by $\delta I_0 = \gamma\delta\nu/\tau = \gamma\sqrt{\nu}/\tau$. Thus, the theoretical detected peak signal-to-noise ratios $\mathrm{SNR}_1$ and $\mathrm{SNR}_2$ of our experimental scheme when displaying measurements $\hat{\Pi}_1$ and $\hat{\Pi}_2$ can be obtained in Eq. (35).

## Data availability

The data that support the findings of this study are available within the paper and its Supplementary Information. Any additional information is available from the corresponding authors upon request.

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

## Acknowledgements

We thank professor Guoyong Xiang for the helpful discussions. B.X. thanks Miaomiao Liu and Zhiyue Zuo for their help in setting up the experiment. The authors are thankful for the financial support from the National Natural Science Foundation of China (Grants No.62071298, No. 61671287, No.61631014, and No. 61901258) and the fund of the State Key Laboratory of Advanced Optical Communication Systems and Networks.

## Author contributions

G.Z. conceived the research project, J.H. designed the scheme, B.X. constructed the theoretical model and carried out the experiments with assistance from J.H., H.L., H.W., and B.X. analyzed the data. B.X. and J.H. wrote the manuscript with contributions from all authors.

## Competing interests

The authors declare no competing interests.
