## [Peer Review File · Nature Communications]

Toward Incompatible Quantum Limits on Multiparameter EstimationREVIEWER COMMENTS

Reviewer #1 (Remarks to the Author):

Due to Heisenberg's uncertainty principle, the optimal measurements for estimating individual parameters may be incompatible. As a result, there exists a tradeoff between estimation errors for different parameters. Recently, the tradeoff relation for quantum multiparameter estimation is given by the reference [26] in the main text of this manuscript. The tradeoff relation for estimating two parameters is achievable for pure states. It is difficult to find and implement the optimal measurements for the practical problem of jointly estimating two parameters. In this manuscript, the authors theoretically design and experimentally implemented a scheme that approaches the fundamental quantum limit of jointly estimating two parameters---the spatial displacement and angular tilt of light.

In this work, the authors synthesize several novel technologies to approach the "quantum limit point". They used the high-order Hermite-Gaussian modes as the probe state. Moreover, they used weak value amplification and spatial mode decomposition to implement the measurement. Therefore, I think this work is significant to the experimental research on quantum metrology.

The manuscript is correct, reasonably well written, and well structured. Both the theoretical derivation and experiment technologies are depicted in enough detail. The results are relevant for researchers working in quantum metrology.

I also have the following comments and suggestions.

- In the Introduction, when the authors mentioned superresolution, they cited the reference [7]: "W.-K. Tham, H. Ferretti, and A. M. Steinberg, Beating rayleigh's curse by imaging using phase information, Phys. Rev. Lett. 118, 070801 (2017)". This reference is an experimental demonstration of the theoretical paper "M. Tsang, R. Nair, and X.-M. Lu, Quantum Theory of Superresolution for Two Incoherent Optical Point Sources, Physical Review X 6, 031033 (2016)". So I suggest that the authors also cited the above-mentioned theoretical paper.

- I think the authors should explain the Hamiltonian of the unitary transformation caused by the PZT.

- In the Supplemental Material, it is an inconsistency between the definition of C_{ij} in Eq. (7) and the inequality of Eq. (8). If $\mathcal{C}_{ij} = C_{ij}$, as claimed by the authors in the second line below Eq. (7), then the term " \mathcal{C}_{ij}^2 " should be " C_{ij}^2 ". This inconsistency does not impact the final result of the inequality expressed in terms of the quantum geometric tensor.

Reviewer #2 (Remarks to the Author):

This paper contains a theoretical part and an experimental part. In the theoretical part, a trade-off bound, which contains a constant related to the uncertainty principle (called QMEC by the authors), is introduced concerning the estimation errors of two parameters in a quantum state. The trade-off bound (which was also derived in a previous work) performs better than the traditional quantum Cramer-Rao bound. Then the authors consider the specific problem of estimating position and momentum shifts in a bosonic system, propose using Hermite-Gaussian states to enhance the sensitivity of estimating the two parameters, calculate the Fisher information and show that the quantum limit is asymptotically achievable when the energy of the state tends to infinity. In the experimental part, the authors demonstrate the experimental results of estimating position and momentum shifts in a bosonic system using the Hermite-Gaussian states and show the quantum limit is approachable.

There are many problems of the presentation and writing in this paper, and I find it hard to consider acceptance without several major revisions. Here are some of my major concerns:

(1) In the theoretical part, the authors use “ δg_i tilde” to characterize the estimation precision of parameter “ g_i ” and make many calculations and figures using this definition (e.g., Eq.(5), Eq.(27), Fig 2, Fig 4, etc.). However, in reality, the correct figures of merit should be the estimation error “ δg_i ”. Moreover, in the experimental part, the plots were made using “ δg_i ”, inconsistent with “ δg_i tilde” in the theoretical part. I think it would make much more sense if the authors can use “ δg_i ” throughout to compare different estimation protocols. Furthermore, the “quantum limits” should not be defined using the artificially introduced functions “ δg_i tilde”.

(2) The authors show advantage of the trade-off bound (Eq.(5)) over the traditional quantum Cramer-Rao bound in the theoretical part. However, there are many other useful bounds for multi-parameter estimation that the authors fail to mention. For example, the Holevo bound (see e.g. PhysRevX.11.011028), which is proven to be achievable asymptotically using collective measurements; or other bounds (see e.g. PhysRevLett.128.250502) for local measurements. It seems strange to only consider the uncertainty principle type of bounds.

(3) One conclusion of the authors is that when n tends to infinity, the quantum limit is asymptotically approachable (although the authors only show it explicitly in Eq.(27) using the artificial “ δg_i tilde” instead of the estimation error “ δg_i ”). However, it is not surprising that the larger the energy is, the preciser the estimation is. Therefore, in order to make the conclusion more convincing, the authors should make a comparison between their results and other quantum states (e.g. Gaussian states) on the same energy level, in order to convince readers of the advantage of the HG states.

Other comments:

(1) It was mentioned many places in the paper that it is “counter-intuitive” that one should choose a state with a large variance. I find it confusing. Even if you just want to estimate a single parameter, it is still better to choose a state with a large variance to improve the estimation precision. So why is it counter-intuitive that for two parameters, large variance is preferred?

(2) This authors did some calculations concerning direct imaging in the theoretical part. I am not sure why that’s relevant.

(3) In Fig. 6, it is extremely unclear the difference between the yellow cross (experiment) and the blue curve (bound). It would be much clearer if the plots are made for each n separately. It would also be better if the theoretical calculations are plotted.

(4) Eq.(59) is quite confusing and doesn’t match the explanatory words before Eq.(59).

(5) If my understanding is correct, the experiment is done when SNR is fixed. Should the sample number n_u be fixed in order to make a fair comparison? It need more clarification.

Possible typo:

(1) In Table I, should it be $SNR_2 = 1$ instead of $SNR_1 = 2$?

1 Responses to reviewer 1

1. Response to comment: *“In the Introduction, when the authors mentioned superresolution, they cited the reference [7]. “W.-K. Tham, H. Ferretti, and A. M. Steinberg, Beating rayleigh’s curse by imaging using phase information, Phys. Rev. Lett. 118, 070801 (2017)”. This reference is an experimental demonstration of the theoretical paper “M. Tsang, R. Nair. and X.-M. Lu, Quantum Theory of Superresolution for Two Incoherent Optical Point Sources, Physical Review X 6, 031033 (2016)”. So I suggest that the authors also cited the above-mentioned theoretical paper.”*

Thanks for your suggestion, we have cited this theoretical paper in the revised manuscript.

2. Response to comment: *“I think the authors should explain the Hamiltonian of the unitary transformation caused by the PZT.”*

Thanks for your suggestion, we have given the unitary transformation caused by the PZT in Eq. (28) of the revised manuscript and explained how the PZT results in simultaneous spatial displacement and momentum kick of light beam in the nearby text.

Experimentally, we generate the displacement and tilt signals for light beam via a PZT driven mirror in a Mach-Zehnder interferometer (MZI), where only the horizontal polarization component ($|H\rangle$) of light beam is modulated, as is shown in the above figure. The mirror is driven by two parallel PZT chips, then the synchronous signals on two chips cause the displacement only for the light beam, and the reverse signals on two chips cause the tilt only for the light beam. In the experiment, we exert the asynchronous signals with relative phase $\Delta = 4^\circ$ on two chips, which gives the tiny spatial displacement d and tiny angular tilt θ for the

light beam simultaneously (the experimental details are given in the section V.A of the revised manuscript). Then the unitary transformation caused by the PZT is given as:

$$\exp^{-i\hat{p}_w \otimes \hat{x} - i\hat{p}_w \otimes \hat{x}}$$

where $\hat{p}_w = \frac{1}{2}(\hat{p}_1 + \hat{p}_2)$, $\hat{x} = 2\hat{x}$ is the wave number of light beam, thus, \hat{p}_w is the corresponding transverse momentum kick for light beam.

The corresponding changes are highlighted and marked as Change #6 in Page 6 of the revised manuscript.

To make the experimental part consistent with the theoretical part, we concern with the parameters \hat{p}_1 and \hat{p}_2 generated by operators \hat{p}_w and \hat{x} when propagation of HG beams is involved. (Here, we give up the artificial parameter \hat{p}_w in the experimental part of the revised manuscript.)

To be clear, the final state of the whole weak measurement system in the theoretical part without -propagation is

Thus, the unitary parameterization in the theoretical part is $|\Psi_w\rangle = |\langle \hat{p}_w | \hat{x} \rangle\rangle = |\langle \hat{p}_w | \hat{x} \rangle\rangle$

$$\hat{p}_w = \hat{p}_1 + \hat{p}_2 = \exp^{-i\hat{p}_1 \otimes \hat{x} - i\hat{p}_2 \otimes \hat{x}}$$

Nevertheless, the final state of the whole system in the experimental part with -propagation is

where $\hat{p}_0 = \hat{p}_1 + \hat{p}_2$, then the unitary parameterization in the experimental part is $|\Psi_w\rangle = |\langle \hat{p}_0 | \hat{x} \rangle\rangle = |\langle \hat{p}_0 | \hat{x} \rangle\rangle$

Thus, the parameters $\hat{p}_1 = \hat{p}_0$ and $\hat{p}_2 = 0$ in the experimental part. The details of derivation are given in the method part of the revised manuscript.

The corresponding changes are highlighted and marked as Chang #3 in Page 3, Change #5 in Page 6, Change #7 in Page 7, and Change #18 in Page 12 of the revised manuscript.

3. Response to comment: “In the Supplemental Material, it is an inconsistency between the definition of \hat{p}_w in Eq. (7) and the inequality of Eq. (8). If $\hat{p}_w = \hat{p}_0$, as claimed by the authors in the second line below Eq. (7), then the term “42” should be “2”. This inconsistency does not impact the final result of the inequality expressed in terms of the quantum geometric tensor.”

We are very sorry for this typo, $\hat{p}_w = \hat{p}_0$ should be $\hat{p}_2 = 0$ here. This typo has been corrected in the revised Supplemental Material.

Finally, thank you for your review and good suggestions.

2 Responses to reviewer 2

1. Response to comment: “In the theoretical part, the authors use \hat{p}_w to characterize the estimation precision of parameter \hat{x} and make many calculations and figures using this definition (e.g., Eq.(5), Eq.(27), Fig 2, Fig 4, etc.). However, in reality, the correct figures of merit should be the estimation error. Moreover, in the experimental part, the plots were made using \hat{p}_0 , inconsistent with \hat{p}_w in the theoretical part. I think it would make much more

sense if the authors can use $\tilde{\epsilon}$ throughout to compare different estimation protocols. Furthermore, the “quantum limits” should not be defined using the artificially introduced functions $\tilde{\epsilon}$.”

Thank you for this comment. The purpose of introducing $\tilde{\epsilon}$ (in Eqs. (5), (10), etc.) is not for characterizing the estimation precision, but for evaluating the discrepancy between the estimating error on multiparameter estimation and its corresponding quantum limit $1/\tilde{\epsilon}$. In this work, our goal is to find a way to make estimation errors for incompatible parameters approach their quantum limits simultaneously. For this purpose, $\tilde{\epsilon} \equiv \tilde{\epsilon} \geq 1$ would be a more suitable figure of merit to characterize how close $\tilde{\epsilon}$ is to the limit, where $\tilde{\epsilon} = 1$ means $\tilde{\epsilon}$ achieves the limit. Moreover, the normalized $\tilde{\epsilon}$ is dimensionless, which mitigates the influence of the units corresponding to different parameters. As we have proven and demonstrated in the theory part and Fig.4 that the precisions can approach the quantum limits simultaneously, it is free to use $\tilde{\epsilon}$ instead of $\tilde{\epsilon}$ in the experimental part for demonstrating the experimental precision with comparison to the theoretical expectations.

To be clear, we would like to explain in follows why using $\tilde{\epsilon}$ is hard for demonstrating the attainability of simultaneous quantum limits. Here, we concern with the incompatible parameters θ_1 and θ_2 generated by the momentum operator \tilde{p} and the position operator \tilde{x} for the n -order HG initial probe $|\psi\rangle = |n\rangle$. Then the parameterized state is $|\psi(\theta)\rangle = \exp[-i\theta_1\tilde{p} - i\theta_2\tilde{x}]|n\rangle$, which leads to the corresponding QFIM be calculated as

$$F_{\theta_1, \theta_2} = \begin{pmatrix} 2(n+1) & 0 \\ 0 & 4n \end{pmatrix}$$

Thus, the quantum limits for parameters θ_1 and θ_2 increase with the mode number n of initial HG probe.

And θ_2 $\frac{1}{4(n+1)}$ the trade-off bound for estimation errors

θ_1 and θ_2 is given as

$$\theta_1 \geq \frac{1}{2(n+1)}, \quad \theta_2 \geq \frac{1}{4(n+1)}$$

To identify the discrepancy between the practical attainable bound of estimation errors δ_1, δ_2 and the corresponding quantum limits, and study how the mode number of initial HG probe improves this discrepancy, we plot the quantum limits and trade-off bounds of parameters δ_1 and δ_2 with HG_{n-1} and HG_n probe in the above figure.

From this figure, we can see that both quantum limit point and trade-off bound are improved within the increasing of mode number n . However, it is hard to distinguish whether the quantum limit point is improved faster than the trade-off bound or the trade-off bound is improved faster than the quantum limit pointer within the increasing of mode number n from the estimation errors δ_1 and δ_2 in the above figure. Thus, we cannot conclude that the attainability of simultaneous quantum limits for incompatible parameters δ_1 and δ_2 is improved with the increasing of mode number n from the trade-off bounds of the estimation errors δ_1 and δ_2 directly. In contrast, by employing the normalized estimation errors $\tilde{\delta}_1$ and $\tilde{\delta}_2$, the attainability of simultaneous quantum limits for incompatible parameters $\tilde{\delta}_1$ and $\tilde{\delta}_2$ is reflected directly by the trade-off bound of normalized parameters $\tilde{\delta}_1$ and $\tilde{\delta}_2$, which is improved with the increasing of mode number n , as is shown in the following figure (Fig. 4b in the manuscript).

Consequently, the normalized estimation error $\tilde{\delta}$ is more efficient than the direct estimation error δ on evaluating the attainability of simultaneous quantum limits for incompatible parameters.

The corresponding changes are highlighted and marked as Change #2 in Page 3 of the revised manuscript.

2. Response to comment: “The authors show advantage of the trade-off bound (Eq.(5)) over the traditional quantum Cramer-Rao bound in the theoretical part. However, there are many other useful bounds for multi-parameter estimation that the authors fail to mention. For example, the Holevo bound (see e.g. PhysRevX.11.011028), which is proven to be achievable asymptotically using collective measurements; or other bounds (see e.g. PhysRevLett.128.250502) for local measurements. It seems strange to only consider the uncertainty principle of bounds.”

Thanks for your suggestion, we have supplemented the comparison of the

Holevo bound and our trade-off bound in section III.B of the revised manuscript. The Holevo bound reveals the tight lower bound of weighted mean errors on multiparameter quantum estimation. For the post-selected measurement scenario with HG pointer in our scheme, the Holevo bound for the incompatible parameters 1 and 2 is

The Holevo bound has been proved attainable for the weighted mean errors via collective measurements. However, the Holevo bound is difficult to completely identify the trade-off curve regarding the attainable precision limits on estimating different parameters. As is shown in the above figure (Fig. 8 in the revised manuscript), the Holevo bound is tangent to the trade-off bound at $\tilde{\sigma}_1 = \tilde{\sigma}_2$, and the curve of trade-off bound is tighter than the Holevo bound. Thus, the trade-off bound is more informative than the Holevo bound.

The corresponding changes are highlighted and marked as Change #16 in Page 9 of the revised manuscript.

3. Response to comment: “One conclusion of the authors is that when n tends to infinity, the quantum limit is asymptotically approachable (although the authors only show it explicitly in Eq.(27) using the artificial $\tilde{\sigma}$ instead of the estimation error). However, it is not surprising that the larger the energy is, the preciser the estimation is. Therefore, in order to make the conclusion more convincing, the authors should make a comparison between their results and other quantum states (e.g. Gaussian states) on the same energy level, in order to convince readers of the advantage of the HG states.”

Theoretically, the attainability of simultaneous quantum limits for incompatible parameters is independent of the energy level of initial probe state, because the

$$4\langle \Delta \tilde{\sigma}_2 \rangle \langle \Delta \tilde{\sigma}_1 \rangle$$

quantum multiparameter estimation criterion (QMEC) = $\tilde{\sigma}(\tilde{\sigma}_1, \tilde{\sigma}_2)$ depends on the uncertainty properties of initial probe regarding the generators $\tilde{\sigma}_1$ and $\tilde{\sigma}_2$ instead of its energy level.

To be clear, there are two situations are involved for explanation: (1) two Gaussian states with different energy levels; and (2) Gaussian state and n -order HG state with the same energy level. Here, we still concern with the simultaneous measurement of incompatible parameters θ_1 and θ_2 generated by the momentum operator \hat{p}_1 and the position operator \hat{x}_2 , where the unitary parameterization is $U(\theta) = \exp(-i\theta_1 \hat{p}_1 - i\theta_2 \hat{x}_2)$. First, considering the time-independent solution $\psi(x)$ for the Schrödinger equation of harmonic oscillators:

$$i\hbar \frac{\partial}{\partial t} \psi = \left[\frac{\hat{p}_1^2}{2m} + \frac{1}{2} m \omega^2 \hat{x}_2^2 \right] \psi$$

For energy level $E = \hbar \omega (2n + 1/2)$, the corresponding eigenket is $|n\rangle = \int dx \psi_n(x) |x\rangle$, which corresponds to the n -order HG state. And $|0\rangle$

that:

$$\langle \Delta | \psi \rangle = \left(\frac{m\omega}{\pi\hbar} \right)^{1/4} \exp\left(-\frac{m\omega}{2\hbar} x^2\right) = \text{Gaussian state with energy level } = \hbar \omega / 2. \text{ It is easy to obtain}$$

$$2\hbar \omega (2n + 1)$$

$$= \hbar \omega (2n + 1)$$

$\psi_n(x) = \left(\frac{m\omega}{\pi\hbar} \right)^{1/4} \frac{1}{\sqrt{2^n n!}} H_n\left(\sqrt{\frac{m\omega}{\hbar}} x\right) \exp\left(-\frac{m\omega}{2\hbar} x^2\right) = \hbar \omega$ **Situation I:** two Gaussian states with different energy levels,

initial probe state

$|1\rangle = |0, 1\rangle$ and initial probe state $|2\rangle = |0, 2\rangle$, $1 < 2$

In this scenario, the energy level of state $|1\rangle$ ($E^{(1)} = \hbar \omega$) is smaller than the

energy level of state $|2\rangle$ ($E^{(2)} = 2\hbar \omega$). Performing the unitary parameterization

$U(\theta) = \exp(-i\theta_1 \hat{p}_1 - i\theta_2 \hat{x}_2)$ on state $|1\rangle$, the corresponding QFIM regarding the parameters θ_1 and θ_2 can be calculated as

$$F(|0, 1\rangle) = \begin{pmatrix} 2\hbar & 0 \\ 0 & \hbar \end{pmatrix}$$

$$U(\theta) = \exp(-i\theta_1 \hat{p}_1 - i\theta_2 \hat{x}_2)$$

Similarly, performing the unitary parameterization

state $|2\rangle$, the corresponding QFIM regarding the parameters θ_1 and θ_2 can be calculated as

$$F(|0, 2\rangle) = \begin{pmatrix} 2\hbar & 0 \\ 0 & 2\hbar \end{pmatrix}$$

It is easy to determine that $(1, |0, 1\rangle) < (1, |0, 2\rangle)$, but $(2, |0, 1\rangle) > (2, |0, 2\rangle)$, which means that increasing the energy level of Gaussian state will improve the quantum precision limit of parameter 1, but reduce the quantum precision limit of parameter 2. However, the attainability of simultaneous quantum limits for parameters 1 and 2 depends on the QMEC ¹², where the Gaussian states $|0, 1\rangle$ and $|0, 2\rangle$ give the same results:

$${}_{12}(|0, 1\rangle) = {}_{12}(|0, 2\rangle) = 1$$

Thus, the curve properties of the trade-off bounds with the Gaussian states $|0, 1\rangle$ and $|0, 2\rangle$ are the same. This result leads to the same attainability of simultaneous quantum limits for parameters θ_1 and θ_2 with the Gaussian states $|0, 1\rangle$ and $|0, 2\rangle$. For illustration, we have plotted the trade-off bounds with the Gaussian states $|0, 1\rangle$ and $|0, 2\rangle$ in the above figures.

Situation II: Gaussian state and order-1 FC state with the same energy level, initial probe state $|1\rangle = |0, 1\rangle$ and initial probe state $|2\rangle = |, 2\rangle$, $\theta_1 = (2 + 1)\theta_2$

In this scenario, the energy level of state $|1\rangle$ ($E^{(1)} = 2\theta_1 \hbar$) is the same as that of state $|2\rangle$ ($E^{(2)} = \theta_2 + \frac{1}{2}\theta_2 \hbar = 2\theta_1 \hbar$). Performing the unitary

parameterization $U = \exp(-i\theta_1 \hat{Q}_1 - i\theta_2 \hat{Q}_2)$ on state $|1\rangle$, the corresponding QFIM regarding the parameters θ_1 and θ_2 can be calculated as

$$F_{\theta}(|0, 1\rangle) = \begin{pmatrix} 2\theta_1 & 0 \\ 0 & 2\theta_1 \end{pmatrix}$$

Similarly, performing the unitary parameterization $U = \exp(-i\theta_1 \hat{Q}_1 - i\theta_2 \hat{Q}_2)$ on state $|2\rangle$, the corresponding QFIM regarding the parameters θ_1 and θ_2 can be calculated as

$$F_{\theta}(|, 2\rangle) = (2 + 1) \begin{pmatrix} 2\theta_2 & 0 & 2\theta_1 & 0 \\ 0 & 2\theta_2 & 0 & 2(2 + 1)\theta_1 \\ 0 & 0 & 2 & 0 \\ 0 & 0 & 0 & 1 \end{pmatrix}$$

$$F_{\theta}(|0,1\rangle) = 1, \quad F_{\theta}(|, 2\rangle) = \frac{1}{(2 + 1)^2}$$

It is easy to determine that $\langle 1, |0, 1\rangle \rangle = \langle 1, |, 2\rangle \rangle$, but $\langle 2, |0, 1\rangle \rangle =$

$\langle 2+1 \rangle^2 \langle 2, |, 2\rangle \rangle$. Therefore, giving the Gaussian state and the HG state on the same energy level, the quantum limits for the Gaussian state $|0, 1\rangle$ and the HG state $|, 2\rangle$ are the same regarding the parameter 1, but the quantum limit for the

HG state $|, 2\rangle$ is $(2 + 1)^2$ -fold more precise than the Gaussian state $|0, 1\rangle$ regarding the parameter 2.

Although the Gaussian state and the HG state are on the same energy level, their

Thus, the attainability of simultaneous quantum limits for the incompatible parameters 1 and 2 with the HG state $|n, 2\rangle$ is improved $(2n + 1)^2$ -fold than that with the Gaussian state $|0, 1\rangle$, though they are on the same energy level. For illustration, we have plotted the trade-off bounds with the Gaussian states $|0, 1\rangle$ and the HG state $|n, 2\rangle$ separately in the above figures.

From above analysis, we can conclude that the QMEC 12 for evaluating the attainability of simultaneous quantum limits is independent of the energy level of initial probe state. The attainability of simultaneous quantum limits depends on the uncertainty properties of the initial probe state regarding the parameters' generators.

We have supplemented this explanation in the supplemental materials and clarified that this improvement of the attainability of simultaneous quantum limits is independent of the energy level of initial probe state in the revised manuscript.

The corresponding changes are highlighted and marked as Change #4 in Page 5 of the revised manuscript, and we have supplemented section II in the Supplemental Material.

4. Response to comment: *“It was mentioned many places in the paper that it is “counter-intuitive”; that one should choose a state with a large variance. I find it confusing. Even if you just want to estimate a single parameter, it is still better to choose a state with a large variance to improve the estimation precision. So why is it counter-intuitive that for two parameters, large variance is preferred?”*

The “counter-intuitive” in our work refers to that the minimal-uncertainty initial probe state ($n = 1$) gives the worst attainability of the simultaneous quantum limits for incompatible parameters, but this attainability will be improved by increasing the initial probe state's uncertainty. This result is indeed similar to the scenario of estimating a single parameter. The precision on estimating a single parameter only depends on the generator's variances $\langle \Delta_{\sim 2}^2 \rangle$, whereas the precision on estimating incompatible parameters and depends on the Heisenberg's uncertainty

relation $\langle \Delta_{\sim 2}^2 \rangle \langle \Delta_{\sim 1}^2 \rangle \geq \frac{1}{4} \langle \{ \tilde{A}, \tilde{B} \} \rangle^2$. However, to avoid making confusing, we have changed the phrase of “counter-intuitive” in the revised manuscript.

5. Response to comment: *“This authors did some calculations concerning direct imaging in the theoretical part. I am not sure why that's relevant.”*

Direct imaging method is a very important technique on estimating the

transverse spatial displacement or the transverse momentum kick of light beams (Phys. Rev. Lett. 102, 173601, Rev. Mod. Phys. 86, 307, Phys. Rev. X 6, 031033). In the weak measurement scheme, direct imaging is the best POVM for estimating the spatial displacement of light beam when weak value is a real number, nevertheless direct imaging is the best POVM for estimating the momentum kick of light beam when weak value is an imaginary number, which had been proved in Phys. Rev. Applied 13, 034023. From Eq. (23) in the manuscript (CFIM of direct imaging method regarding the spatial displacement Δ_1 and momentum kick Δ_2), the practical precision on estimating spatial displacement Δ_1 only or estimating momentum kick Δ_2 only is improved with the mode-number n .

Moreover, the trade-off relation of the estimation errors of the spatial displacement Δ_1 and momentum kick Δ_2 is given as:

$$\frac{1}{(\Delta_1)^2} + \frac{1}{40} \frac{1}{(\Delta_2)^2} \geq \frac{4(n+1)}{2n+1}$$

which is also improved with the mode number n .

However, we focus on the attainability of the simultaneous quantum limits for incompatible parameters. As we analyzed in the section II of the manuscript, direct imaging cannot improve the attainability of the simultaneous quantum limits by employing high-order HG state. Thus, using this example, we would like to clarify that the measurement method who gives improvement on the precision for each parameter via high-order HG state does not necessarily gives improvement on the attainability of the simultaneous quantum limits.

6. Response to comment: *“In Fig. 6, it is extremely unclear the difference between the yellow cross (experiment) and the blue curve (bound). It would be much clearer if the plots are made for each n separately. It would also be better if the theoretical calculations are plotted.”*

Thanks for your suggestion, we have supplemented the experimental results and theoretical predictions in subfigures (Fig. 6b to Fig. 6f in the revised manuscript) for each n separately.

The corresponding changes are highlighted and marked as Change #8 in Page 7 and Change #13 in Page 8 of the revised manuscript.

7. Response to comment: *“Eq.(59) is quite confusing and doesn’t match the explanatory words before Eq.(59).”*

This equation explains how we get the experimental SNR from the detected results of the spectrum analyzer. To make it clearer, we move this data processing procedure and corresponding experimental results to Section IV of the supplemental material with extended explanation.

8. Response to comment: *“If my understanding is correct, the experiment is done when SNR is fixed. Should the sample number nu be fixed in order to make a fair comparison? It need more clarification.”*

Thanks for your suggestion. In our experiment, the sample number n is fixed as $\approx 1.05 \times 10^7$ by projecting the final beam state on the corresponding initial state $|0\rangle$ before exerting signals on the PZT. We have clarified this procedure in the 7th paragraph of section II.C of the revised manuscript, and the details of the corresponding experimental set-up and results are given in the last paragraph of section V.A of the revised manuscript.

The corresponding changes are highlighted and marked as Change #9 and

Change #13 in Page 8 of the revised manuscript.

9. Response to comment: “*In Table I, should it be $SNR_2 = 1$ instead of $SNR_1 = 2$?*”

We are very sorry for this typo, and it has been corrected in the revised manuscript.

The corresponding changes are highlighted and marked as Change #12 in Page 8 of the revised manuscript.

Finally, thank you for your review and good suggestions.

1. Theoretical predictions of the minimal detectable spatial displacement and angular tilt of light beam have been given in Eq. (39) of the revised manuscript. (Change #10 in Page 8)
2. We have supplemented experimental results of the minimal detectable spatial displacement and angular tilt for each experimental HG mode in Tab. I of the revised manuscript. (Change #11 in Page 8)
3. We have corrected the typo “SNR₁ = 2” as “SNR₂ = 1” in Tab. I of the revised manuscript. (Change #12 in Page 8)
4. We have moved the Section “QMEC and quantum geometric tensor” from the method part of the previous manuscript to the discussion part of the revised manuscript, and the section title has been revised as “Geometrical properties of QMEC”. We have also supplemented the description of the geometrical properties of QMEC in this section. (Change #14 and Change #15 in Page 9)
5. We have supplemented a section for comparing the trade-off bound and the Holevo bound in the discussion part of the revised manuscript. (Change #16 in Page 9)
6. We have moved the Section “Technical advantages of weak value” from the method part of the previous manuscript to the discussion part of the revised manuscript. (Change #17 in Page 10)

REVIEWER COMMENTS

Reviewer #1 (Remarks to the Author):

In this new version, the Authors have satisfactorily addressed all issues I had raised in the previous report. I also believe that they have more or less satisfactorily addressed the issues raised by the other referee. I suggest that the paper be published in Nature Communications.

Besides, there is a typo in Eq. (4) that should be corrected before publication. The full stop in the commutator in Eq. (4) should be a comma.

Reviewer #2 (Remarks to the Author):

The authors' thorough response managed to resolve many of my concerns. However, it seems to me that there are still some clarifications the authors need to do in order to provide the readers a better interpretation of their results.

In multi-parameter estimation, the usual figure of merit people use is $\Delta g_1 + \Delta g_2$ or $(\Delta g_1)^2 + (\Delta g_2)^2$. The smaller they are, the higher estimation precision the sensing procedure has. In this paper, however, the authors didn't directly focus on enhancing this estimation precision. Instead, they focused on minimizing the incompatibility of estimating two parameters. Theoretically, the QMEC is a good figure of merit. The larger the QMEC is, the smaller the incompatibility is. Practically, the trade-off bound provided by QMEC is not always attainable with suboptimal measurements. And a good figure of merit would be $\Delta g_1 \text{ tilde} + \Delta g_2 \text{ tilde}$, where $\Delta g_i \text{ tilde}$ is equal to the ratio between the true Δg_i and the maximum Δg_i . The smaller $\Delta g_1 \text{ tilde} + \Delta g_2 \text{ tilde}$ is, the smaller the incompatibility is. One goal of this paper is to show that using HG modes, $\Delta g_1 \text{ tilde} + \Delta g_2 \text{ tilde}$ can approach its lower bound equal to 2.

If my understanding above is correct, there are a lot of improvements the authors should do in terms of presentation.

(1) First, as I also said in my previous report, the authors mentioned in many places that their findings--the larger the variances are, the smaller the incompatibility is--is counter-intuitive/contrary to traditional/common experience. To me, such a statement is imprecise and misleading, because the seeming contradiction is merely a result of using different figures of merit. Traditionally, people consider two observables H_1, H_2 to be the least incompatible if they saturate the uncertainty relation, meaning the estimation errors of H_1 and H_2 are both small; while here the goal is to estimate g_1, g_2 , a completely different task from measuring H_1, H_2 . Even for single-parameter estimation, measuring H_1 with the largest variance provides the minimum estimation error for g_1 ; and I don't see any contradiction or surprise in this fact. Therefore, I suggest the authors to clarify the different definitions of incompatibilities and to not cause any confusion by suggesting their results is contrary to common knowledge.

(2) I think the authors should spend more time to clarify what they meant by "approaching the quantum limits" in the beginning, e.g. by providing a clear mathematical definition of what they meant by incompatibility. Given the evidence provided in this paper, it is fair to say that approaching the quantum limits does not necessarily mean achieving the optimal estimation precision (minimizing $\Delta g_1 + \Delta g_2$). The Refs the authors cited, e.g. [20][21] and [22][23] aim at minimizing the estimation error, which has a different goal from what the authors consider here. I suggest the authors made this clear in the paper. Additionally, in order to demonstrate "applications in quantum metrology" as claimed in the abstract, the authors should provide some evidence that minimizing incompatibility would usually go hand in hand with minimizing estimation errors. For example, the authors can

explain more on their example of Gaussian states, provide other examples or add theoretical evidence.

1 Responses to reviewer 1

1. Response to comment: *“there is a typo in Eq. (4) that should be corrected before publication. The full stop in the commutator in Eq. (4) should be a comma.”*

We are very sorry for this incorrect writing, and have corrected this typo in the revised manuscript.

The corresponding change is highlighted and marked as Chang #3 in Page 2 of the revised manuscript.

Finally, thank you again for your review.

2 Responses to reviewer 2

1. Response to comment: *“First, as I also said in my previous report, the authors mentioned in many places that their findings--the larger the variances are, the smaller the incompatibility is--is counter-intuitive/contrary to traditional/common experience. To me, such a statement is imprecise and misleading, because the seeming contradiction is merely a result of using different figures of merit. Traditionally, people consider two observables H_1 , H_2 to be the least incompatible if they saturate the uncertainty relation, meaning the estimation errors of H_1 and H_2 are both small; while here the goal is to estimate g_1 , g_2 , a completely different task from measuring H_1 , H_2 . Even for single-parameter estimation, measuring H_1 with the largest variance provides the minimum estimation error for g_1 ; and I don't see any contradiction or surprise in this fact. Therefore, I suggest the authors to clarify the different definitions of incompatibilities and to not cause any confusion by suggesting their results is contrary to common knowledge.”*

Thanks for your suggestion. We have omitted the phrase of “counter-intuitive” or “contrary to traditional/common experience” in the revised manuscript to avoid making confusing. And we have clarified the different definitions of the incompatibilities regarding the estimating of parameters g_1 and g_2 and the measurement of observables \hat{H}_1 and \hat{H}_2 via the words following the Eq.(3) in the revised manuscript.

The corresponding change is highlighted and marked as Chang #2 in Page 2 of

the revised manuscript.

2. Response to comment: *“I think the authors should spend more time to clarify what they meant by ‘‘approaching the quantum limits’’ in the beginning, e.g. by providing a clear mathematical definition of what they meant by incompatibility. Given the evidence provided in this paper, it is fair to say that approaching the quantum limits does not necessarily mean achieving the optimal estimation precision (minimizing $\delta g_1 + \delta g_2$). The Refs the authors cited, e.g. [20][21] and [22][23] aim at minimizing the estimation error, which has a different goal from what the authors consider here. I suggest the authors made this clear in the paper. Additionally, in order to demonstrate ‘‘applications in quantum metrology’’ as claimed in the abstract, the authors should provide some evidence that minimizing incompatibility would usually go hand in hand with minimizing estimation errors. For example, the authors can explain more on their example of Gaussian states, provide other examples or add theoretical evidence.”*

Thanks for your suggestions. For multiparameter estimation, such as estimating parameters g_1 and g_2 regarding the parameterized state $|\psi_g\rangle$ ($\mathbf{g} = (g_1, g_2)$) simultaneously, we can determine that their estimation errors must satisfy the QCR inequalities:

$$\begin{cases} \delta g_1 \geq 1/\sqrt{vQ(g_1)} \\ \delta g_2 \geq 1/\sqrt{vQ(g_2)} \end{cases}$$

In this work, ‘‘approaching the quantum limits’’ means minimizing the estimation errors δg_1 and δg_2 simultaneously by saturating the above QCR inequalities, which is a stronger condition than minimizing the weighted estimation error $\delta g_1 + \delta g_2$. As is shown in follows, $\delta g_1 + \delta g_2$ will be minimized if δg_1 and δg_2 are minimized simultaneously.

First, in case of g_1 and g_2 being compatible (such as the scenarios in references [20][21] and [22][23]), the optimal measurement $\hat{\Pi}_{\text{opt}}^{(1)}$ for parameter g_1 and the optimal measurement $\hat{\Pi}_{\text{opt}}^{(2)}$ for parameter g_2 are commutative. Applying these measurements simultaneously on the parameterized state $|\psi_g\rangle$ is able to saturate the QCR inequalities, i.e., the estimation errors δg_1 and δg_2 are minimized simultaneously. In this situation, the quantum limits are simultaneous achievable, minimizing δg_1 and δg_2 to their quantum limits is stricter than minimizing $\delta g_1 + \delta g_2$.

For incompatible parameters g_1 and g_2 , their optimal measurements $\hat{\Pi}_{\text{opt}}^{(1)}$ and $\hat{\Pi}_{\text{opt}}^{(2)}$ are noncommutative, which means that the QCR inequalities cannot be saturated simultaneously via practical measurements on parameterized state $|\psi_g\rangle$. Then the estimation precision is governed by the trade-off relations (Eq.(5) in our manuscript):

$$2 - \frac{1}{v(\delta g_1)^2 Q(g_1)} - \frac{1}{v(\delta g_2)^2 Q(g_2)} + 2\sqrt{1 - \frac{1}{S_{12}}} \\ \times \sqrt{\left[1 - \frac{1}{v(\delta g_1)^2 Q(g_1)}\right] \left[1 - \frac{1}{v(\delta g_2)^2 Q(g_2)}\right]} \geq \frac{1}{S_{12}}$$

then there exists an optimal measurement $\hat{\Pi}_{\text{opt}}^{(12)}$ is able to optimize the estimation

errors δg_1 and δg_2 to saturate the above trade-off relation, which are definitely larger than the corresponding quantum limits $1/\sqrt{\nu Q(g_1)}$ and $1/\sqrt{\nu Q(g_2)}$. However, we find that increasing \mathcal{S}_{12} via optimizing the initial probe state is able to make the optimal estimation errors δg_1 and δg_2 given by the above trade-off relation approach their quantum limits simultaneously, which means that the estimation errors δg_1 and δg_2 are further minimized. In this situation, practical optimal estimation errors δg_1 and δg_2 are larger than their quantum limits $1/\sqrt{\nu Q(g_1)}$ and $1/\sqrt{\nu Q(g_2)}$, then optimizing the initial probe state to make δg_1 and δg_2 approach their quantum limits also minimize the weighted estimation error $\delta g_1 + \delta g_2$.

Consequently, “approaching the quantum limits” dose not conflict with minimizing the estimation error $\delta g_1 + \delta g_2$, and it is even a stricter condition in quantum metrology.

Moreover, the mathematical definition of “incompatibility” in quantum metrology is usually given by the weak commutative condition (Eq.(2) in our manuscript):

$$\text{Im} \left(\frac{\partial \langle \psi_g |}{\partial g_i} \frac{\partial | \psi_g \rangle}{\partial g_j} \right) = \langle [\hat{H}_i, \hat{H}_j] \rangle = 0$$

Where the two parameters g_i and g_j are incompatible when the weak commutative condition is violated. However, the weak commutative condition only reveals the existence or nonexistence of “incompatibility” regarding two parameters, but fails to evaluate how large the “incompatibility” is. In contrast, the quantum multiparameter estimation criterion (QMEC) \mathcal{S}_{ij} derived in our work is efficient to quantify the “incompatibility”. Large QMEC corresponds to small incompatibility, which means that the optimal estimation errors of δg_1 and δg_2 are simultaneously closer to their quantum limits.

As we have clarified that “approaching the quantum limits” is a stronger condition compared with minimizing the estimation error $\delta g_1 + \delta g_2$, the references [20][21] and [22][23] should be able to support our claims, because they all aim at achieving the quantum limit for each parameter simultaneously. Besides, to demonstrate the applications in quantum metrology, we have supplemented some practical examples in the introduction part of the revised manuscript, and given some examples of Gaussian states about quantum communication and superresolution in the conclusion part.

The corresponding change is highlighted and marked as Chang #1 in Page 1 of the revised manuscript.

Finally, thank you for your review and good suggestions.

REVIEWER COMMENTS

Reviewer #2 (Remarks to the Author):

After two rounds of revision, I believe this manuscript still contains several imprecise or even misleading statements. I cannot recommend it for publication.

First, I don't think the authors really understood my previous complaint on their statements on uncertainty relations (which I've mentioned twice already). As a result, the previous wrong statements were still there in the manuscript. For completeness of review, I'll repeat this mistake here again. Below Eq.(3), the authors state that "Traditionally, one may expect the probe state being chosen to saturate Eq. (3) when directly reading the values of observables H_i and H_j , thereby the overall estimation error on H_i and H_j is minimized". This sentence is simply wrong. No one, traditionally, would expect to minimize the estimation error on g_i by minimizing the estimation error on H_i . The observable for estimating g_i is simply different from H_i . For example, if $H_i = \text{Pauli Z}$ and you want to estimate g_i ; you should prepare $|0\rangle+|1\rangle$ and measure Pauli X, which is the exact opposite from measuring Pauli Z.

Secondly, the authors insisted, in their response letter, that the result on "approaching the quantum limits"---minimizing the incompatibility of estimating g_1 and g_2 ---is stronger than minimizing the estimation error $\Delta^2 g_1 + \Delta^2 g_2$. This is a baseless statement. As a result, the authors still chose to state in the manuscript that they've improved previous trade-off relations and contributed in several ways to quantum metrology applications, which are simply exaggerations. In the response letter, the authors argued that because their choice of probe states (roughly speaking) minimizes the ratios between the true estimation error (Δg_i) and the optimal estimation error (Δg_i^{opt}), then $\Delta^2 g_1 + \Delta^2 g_2$ is also minimized. This statement (though possibly true in special cases) is baseless and can be wrong. The reason is that the optimal estimation error Δg_i^{opt} are also functions of probe states. For example, $\Delta g_1 + \Delta g_2$ might be minimized for probe states where Δg_1^{opt} and Δg_2^{opt} are quite small while $\Delta g_1 / \Delta g_1^{\text{opt}}$ and $\Delta g_2 / \Delta g_2^{\text{opt}}$ are relatively large. There is no guarantee that the minimization of the ratios implies the minimization of the estimation error. As I suggested in my previous report, the authors should make it clear that they are minimizing the incompatibility, not the the estimation error. Otherwise, they need to provide convincing evidence that the estimation error is also minimized. Both were not done.

1 Responses to reviewer 2

1. Response to comment: *“First, I don’t think the authors really understood my previous complaint on their statements on uncertainty relations (which I’ve mentioned twice already). As a result, the previous wrong statements were still there in the manuscript. For completeness of review, I’ll repeat this mistake here again. Below Eq.(3), the authors state that “Traditionally, one may expect the probe state being chosen to saturate Eq. (3) when directly reading the values of observables H_i and H_j , thereby the overall estimation error on H_i and H_j is minimized”. This sentence is simply wrong. No one, traditionally, would expect to minimize the estimation error on g_i by minimizing the estimation error on H_i . The observable for estimating g_i is simply different from H_i . For example, if $H_i = \text{Pauli Z}$ and you want to estimate g_i ; you should prepare $|0\rangle$ and measure Pauli X, which is the exact opposite from measuring Pauli Z.”*

Thanks for your suggestion. In the last manuscript, we would like to simply distinguish the situations of directly measuring two incompatible observables and estimating two incompatible parameters. The statement “Traditionally, one may expect the probe state being chosen to saturate Eq. (3) when directly reading the values of observables H_i and H_j , thereby the overall estimation error on H_i and H_j is minimized” refers to the situation of directly measuring observables H_i and H_j instead of estimating parameters g_1 and g_2 . We are aware of this statement is misleading to some extent. This statement had been omitted in the revised manuscript to avoid confusions.

The corresponding change is highlighted and marked as Change #4 in Page 2 of the revised manuscript.

2. Response to comment: *“Secondly, the authors insisted, in their response letter, that the result on “approaching the quantum limits”---minimizing the incompatibility of estimating g_1 and g_2 ---is stronger than minimizing the estimation error $\delta^2 g_1 + \delta^2 g_2$. This is a baseless statement. As a result, the authors still chose to state in the manuscript that they’ve improved previous trade-off relations and contributed in several ways to quantum metrology applications, which are simply exaggerations. In the response letter, the authors argued that because their choice of probe states (roughly speaking) minimizes the ratios between the true estimation error (δg_i) and the optimal estimation error (δg_i^{opt}), then $\delta^2 g_1 + \delta^2 g_2$ is also minimized. This statement (though possibly true in special cases) is baseless and can be wrong. The reason is that the optimal*

estimation error δg_i^{opt} are also functions of probe states. For example, $\delta g_1 + \delta g_2$ might be minimized for probe states where δg_1^{opt} and δg_2^{opt} are quite small while $\delta g_1/\delta g_1^{opt}$ and $\delta g_2/\delta g_2^{opt}$ are relatively large. There is no guarantee that the minimization of the ratios implies the minimization of the estimation error. As I suggested in my previous report, the authors should make it clear that they are minimizing the incompatibility, not the the estimation error. Otherwise, they need to provide convincing evidence that the estimation error is also minimized. Both were not done.”

Thanks for your suggestions. We are aware of not answering this comment appropriately in the last response. Indeed, the reviewer’s concern is reasonable. The statement of “approaching the quantum limits is stronger than minimizing the estimation error $\delta^2 g_1 + \delta^2 g_2$ ” is not rigorous, because we inappropriately used the phrase “approaching the quantum limits” to refer to our method.

Our method of mitigating the incompatibility of parameters g_i and g_j is to choose probe states that can increase the variances of $\langle \Delta \hat{H}_i^2 \rangle$ and $\langle \Delta \hat{H}_j^2 \rangle$ simultaneously to increase the QMEC \mathcal{S}_{ij} (Eq.(4) in the main text). It offers the following two benefits:

1. The incompatibility is mitigated according to the trade-off relation (Eq.(5) in the main text), which means $\delta g_i/\delta g_i^{opt}$ and $\delta g_j/\delta g_j^{opt}$ will get closer to 1;

2. The optimal estimation errors δg_i^{opt} and δg_j^{opt} are both decreased according to the QCR inequalities $\delta g_i \geq \frac{1}{2\sqrt{v(\Delta \hat{H}_i^2)}}$ and $\delta g_j \geq \frac{1}{2\sqrt{v(\Delta \hat{H}_j^2)}}$.

Therefore, in our method, the optimal estimation errors δg_i^{opt} and δg_j^{opt} are getting smaller when decreasing the incompatibility. In this case, approaching these quantum limits absolutely gives smaller estimation errors regarding to the parameters g_i and g_j .

To make it clearer, we compare the estimation errors of the HG_n and HG_{n-1} probe states here, where g_1 is the spatial displacement and g_2 is the momentum shift.

As is shown in this figure, when we choose higher order HG state to decrease the incompatibility, the theoretical optimal estimation errors δg_1^{opt} and δg_2^{opt} (QL points in this figure) are decreased correspondingly, and the practical minimal

estimation errors δg_1 and δg_2 (trade-off bound curves in this figure) are also decreased. Thus, our scheme not only mitigates the incompatibility of estimating parameters g_1 and g_2 , but also gives smaller estimation errors of parameters g_1 and g_2 .

Moreover, this result has been testified by our experiments. As is shown in the following table, when we adopt higher order HG beams to mitigate the incompatibility, the detected precision on the spatial displacement and angular tilt of beam are also improved.

TABLE I. Experimental results

HG modes	Driving Voltages ^a		Parameters ^b			
	SNR ₁ = 1 ^c	SNR ₂ = 1 ^d	$\delta g_{1\text{det}}^{\min}$	$\delta g_{2\text{det}}^{\min}$	$\delta d_{\text{det}}^{\min}$	$\delta \varphi_{\text{det}}^{\min}$
HG ₁	73.10 mV	107.03 mV	1.90 nm	$6.62 \times 10^{-2} \text{ m}^{-1}$	2.94 nm	8.22 nrad
HG ₂	54.53 mV	79.67 mV	1.42 nm	$4.93 \times 10^{-2} \text{ m}^{-1}$	2.19 nm	6.12 nrad
HG ₃	45.59 mV	66.56 mV	1.19 nm	$4.12 \times 10^{-2} \text{ m}^{-1}$	1.83 nm	5.11 nrad
HG ₄	40.01 mV	58.39 mV	1.04 nm	$3.62 \times 10^{-2} \text{ m}^{-1}$	1.60 nm	4.48 nrad
HG ₅	36.12 mV	53.14 mV	0.94 nm	$3.29 \times 10^{-2} \text{ m}^{-1}$	1.45 nm	4.08 nrad

^a Driving voltages (peak-to-peak value) of PZT chips when SNR₁ = 1 and SNR₂ = 1 with different HG modes.

^b These columns are the experimental results of minimum detected values of parameters g_1 , g_2 and g_1 .

^c SNR₁ corresponds to the $\hat{\Pi}_1$ projection measurement, where parameter g_1 can be directly demodulated.

^d SNR₂ corresponds to the $\hat{\Pi}_2$ projection measurement, where parameter g_2 can be directly demodulated.

Consequently, our method of mitigating the incompatibility goes hand in hand with improving the estimation errors. Indeed, “approaching the quantum limits” is misleading, our method--rather than solely "approaching quantum limits" --can also improve the estimation precision. We have clarified this point in the revised manuscript.

The corresponding changes are highlighted and marked as Change #1, Change #2, Change #3 in Page 1, Change #5 in Page 2 and Change #6 in Page 11 of the revised manuscript.

Finally, thank you again for your review and good suggestions.

REVIEWERS' COMMENTS

Reviewer #2 (Remarks to the Author):

The authors have successfully fixed the two issues I raised in the previous report. This paper now presents a well-balanced research work including experimental progress on estimating displacement and angle using HG states with solid theoretical supports. I am happy to recommend the publication of this paper in Nature Communications.